# Reconstitution of prospermatogonial specification in vitro from human induced pluripotent stem cells

Young Sun Hwang [1,5], Shinnosuke Suzuki[2,5], Yasunari Seita[1,3,5], Jumpei Ito [4], Yuka Sakata[1], Hirofumi Aso[4], Kei Sato [4], Brian P. Hermann [2] & Kotaro Sasaki [1✉]

Establishment of spermatogonia throughout the fetal and postnatal period is essential for production of spermatozoa and male fertility. Here, we establish a protocol for in vitro reconstitution of human prospermatogonial specification whereby human primordial germ cell (PGC)-like cells differentiated from human induced pluripotent stem cells are further induced into M-prospermatogonia-like cells and T1 prospermatogonia-like cells (T1LCs) using long-term cultured xenogeneic reconstituted testes. Single cell RNA-sequencing is used to delineate the lineage trajectory leading to T1LCs, which closely resemble human T1-prospermatogonia in vivo and exhibit gene expression related to spermatogenesis and diminished proliferation, a hallmark of quiescent T1 prospermatogonia. Notably, this system enables us to visualize the dynamic and stage-specific regulation of transposable elements during human prospermatogonial specification. Together, our findings pave the way for understanding and reconstructing human male germline development in vitro.

[1] Institute for Regenerative Medicine, Department of Pathology and Laboratory Medicine, University of Pennsylvania Perelman School of Medicine, Philadelphia, PA 19104, USA. [2] Department of Biology, University of Texas at San Antonio, San Antonio, TX 78249, USA. [3] Bell Research Center for Reproductive Health and Cancer, Nagoya, Aichi, Japan. [4] Division of Systems Virology, Department of infectious Disease Control, International Research Center for infectious Diseases, Institute of Medical Science, The University of Tokyo, Tokyo 1088639, Japan. [5] These authors contributed equally: Young Sun Hwang, Shinnosuke Suzuki, Yasunari Seita. ✉email: ksasaki@upenn.edu

The germline is established as primordial germ cells (PGCs), and undergoes a complex cascade of developmental processes, that result in the formation of either spermatozoa or oocytes, depending on the sex-specific signals provided by the gonad[1]. Errors that occur during any of these steps can lead to variety of medical conditions, including infertility and congenital anomalies. A precise understanding of the mechanisms regulating human germ cell (GC) development therefore has significant implications for a broad range of human diseases.

Infertility affects ~10% of reproductive-age couples worldwide, and males are responsible for nearly 50% of these cases. Non-obstructive azoospermia (NOA), characterized by the lack of sperm in ejaculate, affects about 10% of infertile men. The etiology of NOA remains unknown in nearly half these patients[2–4]. The most severe form of NOA, Sertoli cell-only syndrome (SCOS), is characterized by the complete lack of male GCs and likely arises due to disruption of male GC development in the fetal and/or early postnatal stages[5,6].

In mice, male-specific GC development starts when PGCs colonize fetal testes and differentiate into prospermatogonia (also known as gonocytes), which divide several times as multiplying (M)-prospermatogonia (M) before becoming mitotically quiescent as primary transitional (T1)-prospermatogonia (T1). Immediately after the fetus is born, T1 migrate from the center of the seminiferous cords towards the periphery, which is the site of the spermatogonial stem cell (SSC) niche[7–9]. As they migrate, T1 proliferate and differentiate into secondary transitional (T2)-prospermatogonia (T2). T2 are considered the immediate precursors to SSCs, which are the founding population for establishing ongoing spermatogenesis[7,8]. Historically, male GCs at the prespermatogenesis (gonadal) phase have been referred to as PGCs, gonocytes, fetal spermatogonia, or prespermatogonia[8]. In this paper, we employed the nomenclature originally proposed by Hilscher et al. (M-, T1-, and T2-prospermatogonia), which clearly conveys GC identity in both sex and timing of development[8,10]. We also refer to all pre-gonadal phase (migratory and pre-migratory) GCs as PGCs.

In a study on humans, Li et al.[11] performed single-cell RNA-seq on fetal testicular cells and identified two distinct GC types, mitotic and mitotic-arrest fetal GCs (FGCs). These GC types appear to correspond to M and T1 in mice. Interestingly, in contrast to mice, the transition from M to T1 in humans occurred asynchronously during the first and second trimester[11]. Moreover, unlike murine T1, which predominantly localize within the tubular lumen, at least some of the M and T1 in both humans and non-human primates already reside on the basement membranes of seminiferous cords[11–13]. Such spatiotemporal heterogeneity is unique to primates and suggests species-specific divergence of prospermatogonial specification.

Given such divergence, new approaches that directly address the mechanisms of human male GC development are required to further our understanding of this process and provide species-specific insights into the causes of human male infertility. However, the development of these approaches has been hampered by the ethical and technical constraints in studying human fetal GCs, which are scarce and do not persist into adulthood. A reliable in vitro reconstitution method that accurately recapitulates the fetal phases of human male GC development would allow for scalable expansion and for the manipulation and visualization of developing male GCs, thereby providing the ability to investigate the molecular mechanisms of human male GC development. To this end, we previously established robust in vitro methods to induce human induced pluripotent stem cells (hiPSCs) into human PGC-like cells (hPGCLCs), which resemble pre-migratory stage PGCs[14]. More recently, researchers have reported the successful maturation of female hPGCLCs into pre-meiotic oogonia-like cells using xenogeneic reconstituted ovaries (xrOvaries)[15]. However, approaches for inducing hPGCLC differentiation into prospermatogonia have not been reported. In this study, we established an analogous culture method, which we termed the xenogeneic reconstituted testis (xrTestis) culture, which allows for the differentiation of hPGCLCs into T1 prospermatogonia-like cells (T1LCs). Notably, T1LCs generated under these conditions bear a transcriptome that closely resembles the transcriptome of T1 in vivo. Overall, our culture method accurately recapitulates in vivo human male GC development and allows us to understand the genetic pathways governing this process.

## Results

**Characterization of prospermatogonia in human fetal testes at second trimester.** To validate our in vitro reconstitution of human male GC development, we first required a precise understanding of the lineage trajectory of male GCs in vivo. During the 2nd trimester, human fetal testes consist of heterogenous cell types at different developmental stages, including M and T1[11,16]. We therefore profiled human fetal testes obtained from three donors (Hs31, Hs26, and Hs27) with gestational ages of 17w3d, 18w0d, and 18w5d, respectively (Supplementary Fig. 1a–e). Histologic sections revealed compact, cylindrically shaped seminiferous cords embedded in the highly cellular stroma (Supplementary Fig. 1a). Seminiferous cords showed scattered GCs with large vesicular nuclei and prominent nucleoli. Stroma contained many cells with round nuclei and abundant eosinophilic cytoplasm, features characteristic of fetal Leydig cells (FLCs)[13].

We determined the lineage trajectory of these cells by performing single-cell RNA-seq (scRNA-seq) using a 10x Genomics platform. Of the ~18,000 cells for which transcriptomes were available, 16,429 remained for downstream analysis after removing low-quality cells. These cells contained a median of ~1900–2700 genes/cell at a mean sequencing depth of 22–69k reads/cell (Supplementary Fig. 1b). By profiling the expression of known marker genes in a t-distributed stochastic neighbor embedding (tSNE) plot[11,17–19], we identified clusters representing multiple known fetal testis cell types including $DND1^+$ GCs, $SOX9^+$ Sertoli cells (SCs), $INSL3^+/CYP17A1^+$ fetal Leydig cells (FLCs), $TCF21^+$ stromal cells (ST, also described as Leydig precursors[11]) and $KDR^+$ endothelial cells (ECs) (Supplementary Fig. 1c, d). We also identified minor cell types that were relatively uncharacterized in human fetal testis, including $HLA$-$DRB1^+$ macrophages (MΦ) and $MYH11^+$ smooth muscle cells (SMCs)[20,21].

To identify marker genes characteristic of the annotated clusters, we tested for differentially expressed genes (DEGs) by comparing each cluster with the remaining clusters (Supplementary Fig. 1e, Supplementary Dataset 1). GCs contained the highest number of DEGs, consistent with their unique biological characteristics. These DEGs were enriched in genes bearing the Gene Ontology (GO) terms "DNA repair" and "spermatogenesis" (Supplementary Dataset 1). DEGs in Leydig cells were enriched in genes for cholesterol and steroid biosynthesis, suggesting that these cells may produce androgens[22]. DEGs in ST were enriched for the GO terms "extracellular matrix" and "collagen catabolic process," which may indicate their role in scaffolding testicular tissue architecture (Supplementary Dataset 1). Additional marker genes identified by DEG analysis, and the GO terms enriched for those DEGs are shown in Supplementary Fig. 1d, e and Supplementary Dataset 1.

We next performed clustering analysis for only GCs, which revealed two distinct clusters (Fig. 1a). One cluster expressed known markers for M (mitotic FGCs) and PGCs (migrating FGCs), such as POU5F1, NANOS3, and TFAP2C (Fig. 1b)[11,23,24].

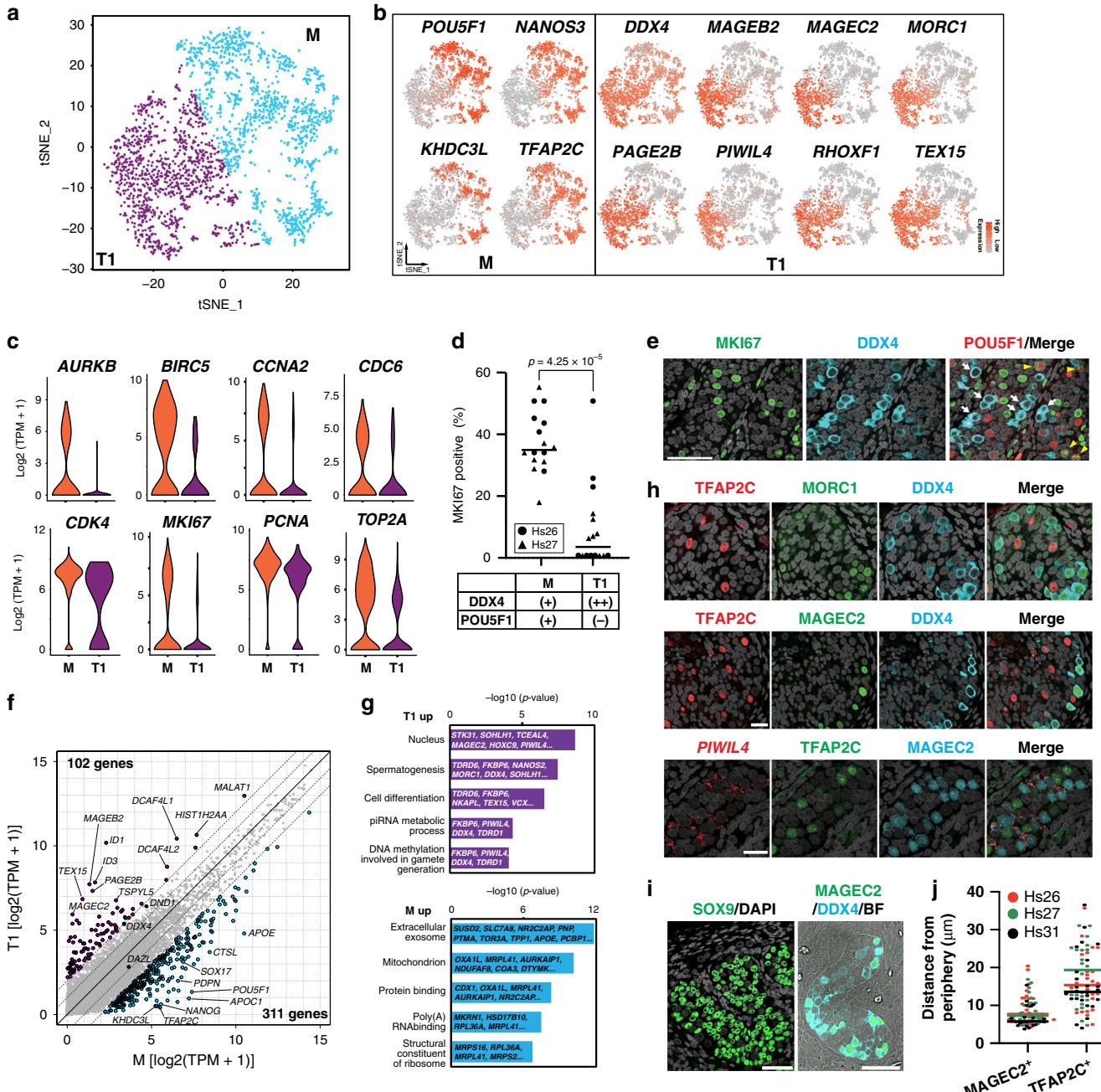

**Fig. 1 Characterization of germ cells in human fetal testis during the second trimester. a** tSNE plot of germ cells (GCs) defined in Supplementary Fig. 1c. Cluster identities were determined by projecting the expression of marker genes onto the same tSNE plot in (**b**). M, M-prospermatogonia (cyan); T1, T1-prospermatogonia (purple). **b** tSNE feature plots of known and newly identified markers for M and T1. **c** Violin plots of proliferation markers for M (1,419 cells) or T1 (1170 cells) defined in (**a**) and (**b**). FDR < $10^{-65}$ for all markers. **d** Percentages of MKI67+ cells among DDX4+POU5F1+ M or DDX4++POU5F1− T1 present in Hs26 (18w0d, black circle) or Hs27 (18w5d, black triangle) samples as assessed by immunofluorescence (IF) analysis. Each dot represents the count per tubular cross section (8 cross sections per sample). P-values for the comparison is determined by two-sided Fisher's exact test ($p = 4.25 \times 10^{-5}$). **e** IF images of a paraffin section of fetal testis (Hs27) targeted for MKI67 (green) and DDX4 (cyan), merged with DAPI (white) and/or POU5F1(red). White arrows indicate DDX4++POU5F1− T1 and yellow arrowheads indicate DDX4+POU5F1+ M co-expressing MKI67. Scale bar, 50 μm. **f** Scatter plot comparing averaged gene expression levels between M and T1 (top). Cyan, genes higher in M; purple, genes higher in T1 (more than fourfold difference [flanking diagonal lines], mean log2[TPM+1] > 2, FDR < 0.01). Key genes are annotated and the number of DEGs are indicated. **g** Gene Ontology (GO) analysis of the DEGs defined in (f). P-value (one-sided Fisher-exact test) and representative genes assigned to each GO term are shown. **h** IF images of paraffin sections of Hs31 for TFAP2C (red) and DDX4 (cyan) combined with MORC1 (green, top) or MAGEC2 (green, middle), and images for combined in situ hybridization (co-ISH) for PIWIL4 (red) with IF for TFAP2C (green) and MAGEC2 (cyan) (bottom). All images are merged with DAPI (white). Merged images for all four color channels are shown at far right. Scale bars, 25 μm. **i** IF images of paraffin sections of Hs26 for SOX9 (green) merged with DAPI (left) or for MAGEC2 (green) and DDX4 (cyan) merged with bright field (BF) (right). IF for SOX9 and BF highlight the border between tubules and the stroma. Scale bars, 50 μm. **j** Distances (μm) from the periphery of tubules for TFAP2C+MAGEC2− M or TFAP2C−MAGEC2+ T1 as quantified by IF images for Hs26 (red), Hs27 (green), and Hs31 (purple). Bars indicate the median value for each cell type per sample. n = 78 (Hs26, 28 cells; Hs27, 30 cells; HS31, 20 cells) and 75 (Hs26, 27 cells; Hs27, 22 cells; HS31, 26 cells) for TFAP2C−MAGEC2+ T1 and TFAP2C+MAGEC2− M, respectively. See also Supplementary Fig. 1 and Supplementary Dataset 1.

The other cluster expressed markers for T1 (mitotic-arrest FGCs), such as *PIWIL4*, *TEX15*, and *RHOXF1*[11,25]. *DDX4* expression was also upregulated in this cluster, which is consistent with the previous immunofluorescence (IF) studies that used DDX4 as a marker for human T1[11] although weaker expression was also seen in M (Fig. 1b). Our IF studies supported the transcriptome clustering results, showing two cell populations within the seminiferous cords, POU5F1+DDX4+ (388/853, 45.5%) and POU5F1−DDX4+/++ (465/853, 54.5%) cells, that represent M and T1, respectively (Fig. 1d, e). T1 exhibited significantly lower transcript levels for proliferation markers, such as *AURKB*, *CCNA2*, *MKI67*, and *TOP2A*, and showed marked reduction of MKI67 protein expression by IF (Fig. 1c, d, e), confirming that these T1 were indeed at the mitotic-arrest stage. RNA velocity analysis using nascent transcripts further corroborated the overall lineage trajectory from M to T1 (Supplementary Fig. 1f)[26].

We examined gene expression patterns across the M-to-T1 transition and found that GC specifier genes (*SOX17*, *TFAP2C*, *PRDM1*, *SOX15*, *NANOS3*) and pluripotency-associated genes (*POU5F1*, *NANOG*, *TCL1B*, *TFCP2L1*) were sharply downregulated as M differentiated into T1 (Fig. 1f, Supplementary Dataset 2). SOX2 was not expressed in either cell types[23,24]. The 311 DEGs identified in M were enriched for the GO terms "mitochondrial inner membrane" and "mitochondrial respiratory chain complex I assembly," suggesting that, as in previous studies on mice, oxidative phosphorylation may be activated in M and downregulated as cells differentiate into T1[27]. The 102 DEGs upregulated during the M-to-T1 transition included X-linked cancer-testis antigens belonging to the MAGE and PAGE gene families, including *MAGEA4*, *MAGEB2*, *MAGEC2*, *PAGE1*, *PAGE2*, and *PAGE2B*. Many genes previously recognized as markers for prospermatogonia (*RHOXF1*, *NANOS2*, *DDX4*) or adult spermatogonia (*SIX1*, *DCAF4L1*, *PLPPR3*, *EGR4*) were also upregulated during this transition[11,25,28]. GO terms in these DEGs correspondingly included "spermatogenesis" (Fig. 1g). Notably, the majority of genes involved in piRNA pathways (e.g., *PIWIL4*, *TEX15*, *MORC1*), which are key guardians of genomic integrity during spermatogenesis, were highly upregulated in T1 (Fig. 1g). Other T1 marker genes in our DEG analysis, such as *MORC1* and *MAGEC2*, have not been previously recognized as markers of human T1.

IF analysis revealed discrete nuclear immunoreactivity for MORC1 and MAGEC2 only in peripherally located DDX4+ T1 (Fig. 1h–j), whereas TFAP2C exclusively marked centrally located M. In situ hybridization (ISH) analysis showed that signals for *PIWIL4* were localized to the perinuclear regions of MAGEC2+ T1 (Fig. 1h). Overall, these findings clearly delineated M and T1 as two distinct male GC types in human fetal testes, each with distinct patterns of gene and protein expression.

**Establishment of male hiPSCs bearing the *TFAP2C-2A-EGFP (AG)*; *DDX4/hVH-2A-tdTomato (VT)*; *PIWIL4-2A-ECFP (PC)* alleles (9A13 AGVTPC).** Using the information from our high-resolution transcriptomic characterization of prospermatogonial development, we attempted to reconstitute this process in vitro using hiPSCs as our starting material. Our transcriptomic analysis, coupled with previous reports in humans and non-human primates, indicated that *DDX4* and *PIWIL4* expression marks T1 and that the expression of both genes is maintained at least until spermatogenesis commences[11,12,29]. *DDX4* expression is likely upregulated earlier than *PIWIL4* given the weaker but significant expression of *DDX4* in M (Fig. 1b)[11]. In addition, *TFAP2C*, a marker for PGCs, was swiftly downregulated upon differentiation into T1 (Fig. 1b, f, h). These observations led us to hypothesize that a combination of *TFAP2C*, *DDX4*, and *PIWIL4* would serve as a powerful marker for visualizing the transition from hPGCLCs to the prospermatogonial stage.

To this end, we introduced targeted *DDX4/hVH-2A-tdTomato* (VT) and *PIWIL4-2A-ECFP* (PC) alleles into previously established *TFAP2C-2A-EGFP* (AG) hiPSCs (585B1 1-7, XY)[14] to generate hiPSCs bearing triple knock-in fluorescence reporters (AGVTPC) (Supplementary Fig. 2a–g). One clone, 9A13, demonstrated successful biallelic targeting of both VT and PC (Supplementary Fig. 2c, d). 9A13 hiPSCs could be stably maintained under feeder-free conditions and exhibited a normal male karyotype (46, XY) (Supplementary Fig. 2e). They formed round, tightly packed colonies, characteristic of hiPSCs (Supplementary Fig. 2f), and expressed the pluripotency-associated markers, POU5F1, SOX2, and NANOG (Supplementary Fig. 2g). We also confirmed that 9A13 hiPSCs were able to differentiate into hPGCLCs through incipient mesoderm-like cells (iMeLCs) with an induction efficiency of ~53% of AG+ hPGCLCs (Supplementary Fig. 2h, i, j, k), consistent with a previous study[14].

**Establishment of xrTestis.** A previous study successfully reconstituted mouse fetal prospermatogonia from mESC-derived PGC-like cells (mPGCLCs) using reconstituted testes, in which dissociated mouse fetal testicular somatic cells were mixed with mPGCLCs before culture[30]. As a first step in applying this methodology to humans, we examined whether dissociated cells from mouse fetal testes could be reassembled in the absence of mouse(m)PGCs or mPGCLCs (Fig. 2a).

E12.5 mouse fetal testicular cells depleted of mPGCs readily formed tight aggregates, or reconstituted testis (rTestis), after floating culture for 2 days (Fig. 2b). After an additional 14 days of culture on an air-liquid interface (ALI), rTestes exhibited numerous anastomosing tubular structures comprised of SOX9+ SCs surrounded by ACTA2+ peritubular myoid cells (Fig. 2c, d, e). Neither TFAP2C+ nor DDX4+ GCs were present, indicating the successful depletion of mPGCs (Fig. 2d). Tubules were surrounded by NR2F2+ stroma reminiscent of mouse fetal testis, but CD31+ endothelial cells and HSD3B1+ Leydig cells were not observed (Fig. 2e).

We then identified culture conditions that would allow hPGCLCs to be integrated into rTestes while maintaining testicular tissue integrity and maximizing the hPGCLC recovery after 14 days of ALI culture. Floating aggregates cultured using Knockout Serum Replacement (KSR)-based media formed larger aggregates with less surrounding cell debris and enhanced the recovery of live cells compared with those cultured in fetal bovine serum (FBS)-based media (Fig. 2f, g). The addition of Y-27632, a potent inhibitor of Rho-associated-coiled-coil containing protein kinase (ROCK), during floating culture further enhanced the formation of tighter, slightly larger aggregates and increased the recovery of AG-positive cells (Fig. 2f, g)[31]. Moreover, IF analysis revealed that aggregates cultured for 14 days in ALI culture using KSR- but not FBS-based medium formed distinct tubules that integrated AG+TFAP2C+ hPGCLC-derived cells (Fig. 2f, h, i, j). IF analysis also confirmed that hPGCLCs maintained PGC markers (POU5F1, NANOG, SOX17, and TFAP2C) after 14 days of ALI culture (Fig. 2i). All GCs were uniformly labeled by AG and by a human mitochondrial antigen (hMito), suggesting that they were derived from hPGCLCs and not from endogenous mouse PGCs (Fig. 2h–j). Leydig cells and endothelial cells were not readily observed in aggregates, suggesting that these cell types might have different culture requirement for survival and/or proper differentiation (Fig. 2j). Overall, these findings confirmed that dissociated mouse testicular somatic cells and hPGCLCs can self-assemble to form mini-fetal testicular tissues, which we named xenogeneic reconstituted testis (xrTestis).

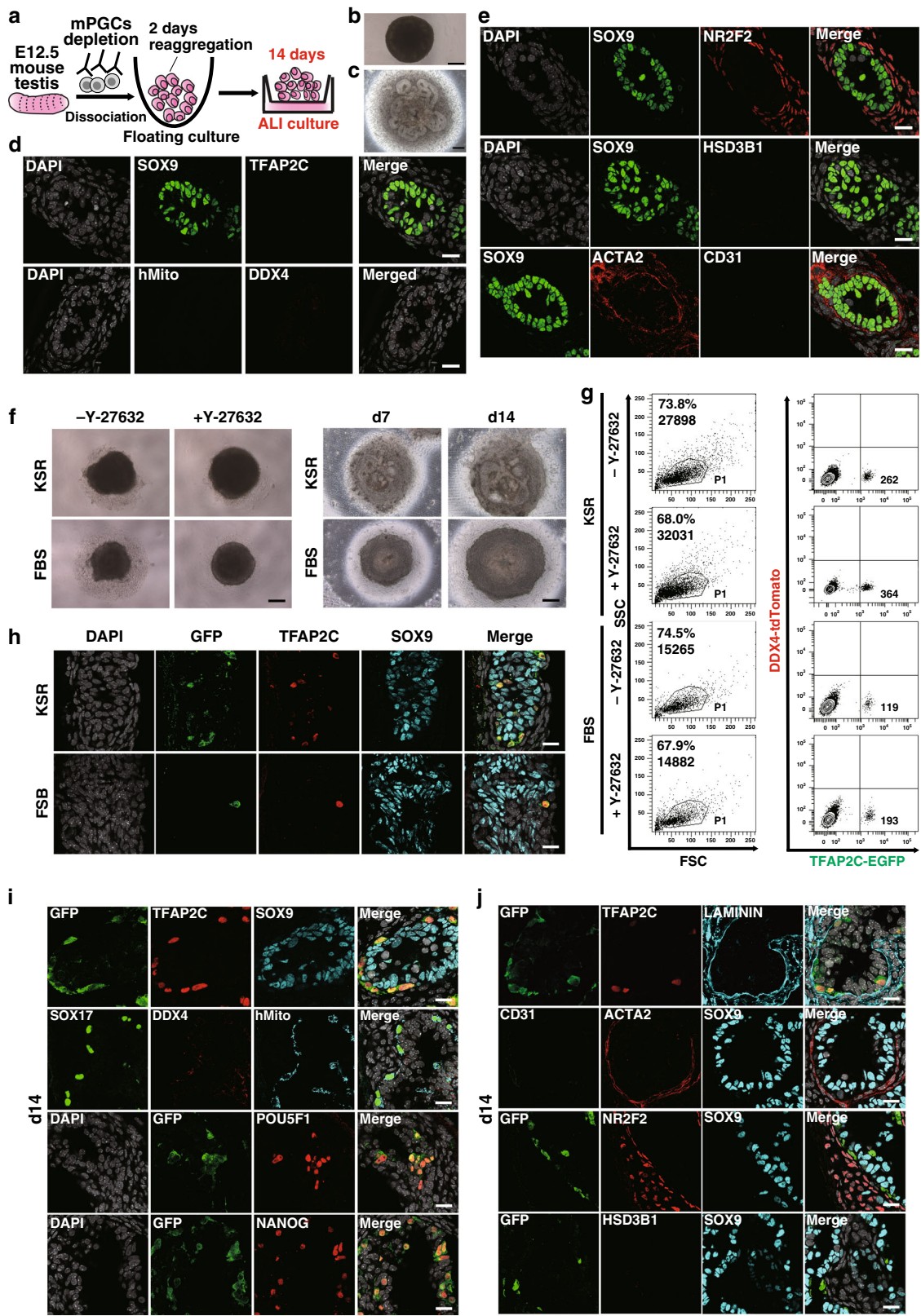

**Extended culture of xrTestis.** We cultured xrTestes for a prolonged period to determine if hPGCLCs could be further differentiated into more advanced male GCs (Fig. 3a). IF analysis of xrTestis cultured for 42 and 77 days revealed the persistence of tubular structures consisting of SOX9+ SCs and AG+ hPGCLC-derived cells that remained predominantly localized within tubules (Supplementary Fig. 3a, b). Although essentially all GCs in 42-day (d42) xrTestes showed an early PGC phenotype (AG+/TFAP2C+/DDX4−/DAZL−), many of the AG+TFAP2C+ GCs in d77 xrTestes were strongly immunoreactive for the M marker, DAZL and somewhat more weakly reactive for DDX4 and VT (Fig. 3b, c, d, Supplementary Fig. 3b, c). These cells exhibited

**Fig. 2 Optimization of rTestis and xrTestis culture. a** Scheme for rTestis culture using mouse testicular somatic cells at embryonic day (E) 12.5. Mouse PGCs (mPGCs) were depleted by MACS. ALI, air-liquid interphase; rTestis, reconstituted testis. **b, c** Bright-field (BF) images of d2 floating aggregate (**b**) and d14 xrTestis on ALI culture (**c**). Scale bars, 200 μm. **d** Immunofluorescence (IF) images of rTestes at d14 for GC markers (red: TFAP2C, DDX4), the human-specific marker human mitochondrial antigen (hMito) (cyan), and the Sertoli cell marker SOX9 (green), with merges with DAPI (white). Scale bars, 20 μm. **e** IF images of rTestes at d14 for somatic cell markers (green: SOX9; red: NR2F2, HSD3B1, ACTA2; cyan: CD31) with merges with DAPI (white). Scale bars, 20 μm. **f** BF images of d2 floating aggregates (left), and d7 and d14 xrTestes (right) cultured in KSR-based or FBS-based medium. Floating aggregates are cultured in the presence or absence of Y-27632 (left). Y-27632 is included in all xrTestis culture (right). Scale bars, 200 μm. **g** FACS analysis of d2 floating aggregates cultured in KSR or FBS-based medium to assess the number of total cells (in dot plot showing SSC [side scatter] and FSC [forward scatter], left) and the number of hPGCLC-derived cells (TFAP2C-EGFP [AG]-positive, right). The percentages of cells in P1 gates (living cells), the total cell numbers (left), and the numbers of AG-positive cells per floating aggregate (right) are shown. **h** IF images of d14 xrTestes cultured in KSR- or FBS-based medium for GFP (green), TFAP2C (red), SOX9 (cyan), and DAPI (white) with their merges. Scale bars, 20 μm. **i** IF images of d14 xrTestes for GC markers (red: TFAP2C, DDX4, SOX17, POU5F1 or NANOG), markers for hPGCLC-derived cells (green: GFP; cyan: hMito), a Sertoli cell marker (cyan: SOX9), and DAPI (white), with their merges. Scale bars, 20 μm. Note that DDX4 is not expressed in xrTestes at this stage. **j** IF images of d14 xrTestes for a GC marker (red: TFAP2C), a basement membrane marker (cyan: LAMININ), and somatic cell markers (green: CD31; red: NR2F2, ACTA2, HSD3B1; cyan: SOX9), with their merges with DAPI (white). Scale bars, 20 μm. See also Supplementary Figs. 2 and 3.

slightly larger and more euchromatic nuclei with low DAPI intensity and prominent nucleoli, characteristic of primate PGCs (Fig. 3b)[12]. IF analysis targeting 5-methylcytosine (5mC) revealed that these hPGCLC-derived cells consistently exhibited low levels of global DNA methylation (Supplementary Fig. 3d). Furthermore, these cells also retained pluripotency-associated markers, such as POU5F1 and NANOG (Fig. 3d, e), suggesting that most hPGCLC-derived cells had progressed towards M but had not yet differentiated into the T1. Flow cytometry analysis at d81 showed the emergence of AG⁺VT⁺ and AG⁻VT⁺ GCs within the dissociated xrTestes, representing 7.9 and 1.7% of total cells (62% or 13% of all human GCs), respectively (Fig. 3f).

We extended our ALI culture further and found that, at d120, xrTestes showed scattered DDX4⁺/DAZL⁺/VT⁺ cells with markedly reduced reactivity for AG, TFAP2C, POU5F1, and NANOG, suggesting that these cells had further differentiated beyond the M stage (Fig. 3c, d, e). We also noted strong expression of the T1 markers MAGEA3 and MAGEC2 at d120 (Fig. 3g)[32]. Flow cytometric analysis confirmed that most GCs had progressed to the AG⁻VT⁺ fraction (4.2% of total cells, 58% of all human GCs) (Fig. 3f). Moreover, some of these AG⁺VT⁺ (14.3%) and AG⁻VT⁺ cells (16.5%) expressed PC, another discriminating marker of T1 (Fig. 3h). IF analysis of xrTestes at d120 showed a marked reduction of MKI67⁺ GCs compared to d77 xrTestes, consistent with progression into the mitotically arrested T1 state (Fig. 3i, j). IF for 5mC revealed that the DNA methylation levels of DDX4⁺ GCs in d120 xrTestes were still lower than those in somatic cells, thus suggesting that overt de novo DNA methylation might not have commenced yet (Fig. 3k). Together, these findings support our conclusion that hPGCLCs differentiated into T1LCs after ALI culture with xrTestes.

**Lineage trajectory leading to T1LCs.** To fully capture the male germ lineage progression without the bias introduced by our a priori markers, we isolated GCs (AG⁺ or VT⁺) from xrTestes at d81 and d124 and evaluated their transcriptomes by scRNA-seq. Precursor cell types (hiPSCs, iMeLCs and hPGCLCs) were also examined by scRNA-seq. Between 344-1,251 cells for each population were used after excluding results from low-quality cells (Supplementary Fig. 4a). We detected ~3000–7000 median genes/cell at a mean sequencing depth of between 68-273k reads/cell (Supplementary Fig. 4a). After computationally aggregating high-quality cells and presenting the results in the same tSNE space (Fig. 4a, Supplementary Fig. 4b, c), we identified four main clusters (clusters 1–4) that largely separated by sample type, as expected (Supplementary Fig. 4b, g, h). Cluster annotation was further validated by the expression of the known markers for each cell type such as SOX2 (marker for hiPSCs and iMeLCs), EOMES

(marker for iMeLCs), and NANOS3 (marker for hPGCLCs and later stage GCs) (Supplementary Fig. 4b)[14,15].

Cluster 4 consisted of GCs from both d81 and d124 xrTestes that expressed DAZL, DND1, PIWIL2, which was consistent with their prospermatogonial status (Supplementary Fig. 4b, g). Further analysis for only cells in cluster 4 revealed three distinct subclusters (4-1, 4-2, and 4-3; Supplementary Fig. 4c). Markers for M that were defined previously in vivo, such as KHDC3L and POU5F1, were uniquely expressed in cluster 4-1. We therefore, identified cells from this cluster as "M-prospermatogonia-like cells" (MLCs). Markers for T1, such as MAGEC2 and PIWIL4 were exclusively expressed in cluster 4-3, suggesting that this cluster represented T1LCs (Supplementary Fig. 4c). Cluster 4-2 was located between MLCs and T1LCs, expressing unique markers, ASB9 and FZD8 (Supplementary Fig. 4c). We posit that these cells were transitional cells (TCs) because of their position in the tSNE plot and their intermediate expression levels of both MLC (POU5F1, NANOS3) and T1LC markers (TEX15) (Supplementary Fig. 4c). Consistent with this interpretation, strong ASB9-positive cells co-expressing intermediate levels of M and T1 markers were also seen at the M/T1 border in the tSNE plot for in vivo human testicular GC samples presented above, although these in vivo cells were not identified as a distinct cluster in tSNE analyses performed with higher clustering resolution (K-means up to 10) (Supplementary Fig. 4d, e).

To understand the lineage trajectory among cells in cluster 4, we performed RNA velocity analysis after re-clustering (Fig. 4b and Supplementary Fig. 4f). This analysis confirmed that a lineage progression from POU5F1⁺ MLCs to PIWIL4⁺ T1LCs occurred in xrTestis cultures (Fig. 4b). Corresponding to this progression, the proportion of cells derived from d124 xrTestis in each cluster increased from 10.2% in the MLC cluster to 74.1% in the TC cluster and 99.3% in the T1LC cluster (Supplementary Fig. 4h).

We also noted that two biological replicates for hPGCLCs (hPGCLCs_1, hPGCLCs_2) formed a single cluster in the tSNE plot (Supplementary Fig. 4g) and showed high concordance of both averaged gene expression using whole-genome data (r2 = 0.98) (Supplementary Fig. 4i) and pairwise DEGs between hPGCLCs and MLCs (Supplementary Fig. 4j). Dissociation-induced genes were not over-represented in any of the identified clusters, ruling out the possibility that clusters were made by any dissociation-induced artifacts (Supplementary Fig. 5a-c)[33]. These observations underscore the robustness of the present scRNA-seq platform.

To more fully understand the molecular mechanisms driving the progression from hPGCLCs through MLCs to T1LCs, we explored the gene expression dynamics of our in vitro-derived GCs. The core pluripotency-associated genes, POU5F1 and NANOG, along

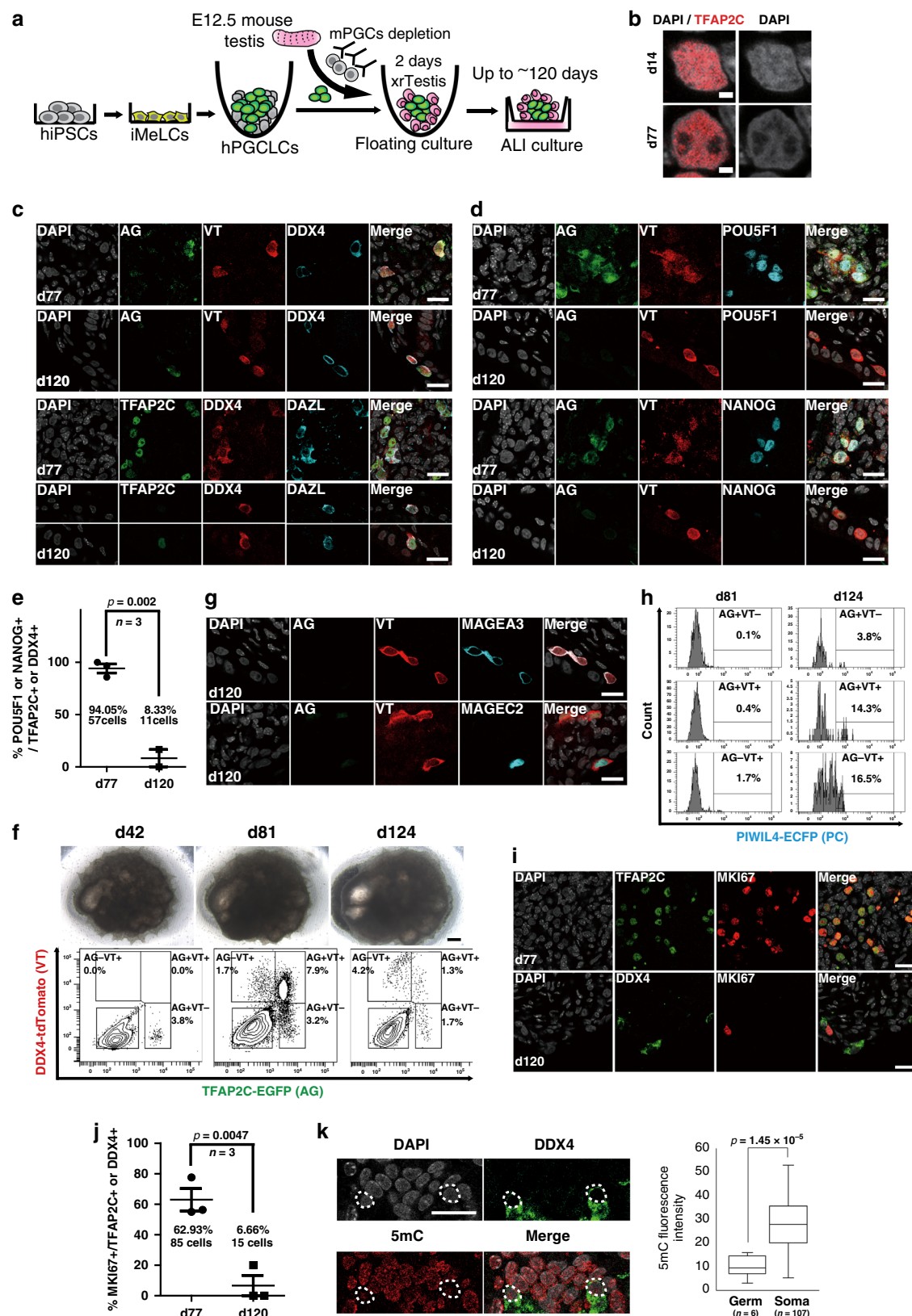

with GC specifier genes (*PRDM1, SOX17, TFAP2C, SOX15*), showed peak expression levels at the hPGCLC stage and declined thereafter (Fig. 4c). Interestingly, the expression of some genes involved in naïve pluripotency, such as *TCL1B, TFCP2L1,* and *ZFP42,* peaked at the MLC stage. Prospermatogonial (or oogonial)

markers, such as *DAZL, DDX4,* and *DPPA3* emerged in MLCs and persisted in later GCs (Fig. 4c)[11,23], which was consistent with our IF and flow cytometric data. Moreover, proliferation markers such as *CCNA2, CDK4, MKI67,* and *TOP2A* were significantly down-regulated in T1LCs (Fig. 4d), which was also consistent with IF

**Fig. 3 Establishment of xenogeneic reconstituted testis (xrTestis) and generation of human T1LCs. a** Scheme for T1LC induction by xrTestis culture. ALI, air-liquid interphase. **b** IF images of hPGCLC-derived cells in d14 and d77 xrTestes for DAPI (white) and TFAP2C (red) with their merges. Scale bars, 2 μm. **c** IF images of d77 or d120 xrTestes for their expression of indicated key GC markers (cyan: DAZL; green: TFAP2C, TFAP2C-EGFP [AG]; red: DDX4, DDX4-2A-tdTomato [VT]) and DAPI (white). Merged images are shown on the right. Scale bars, 20 μm. **d** IF images of d77 or d120 xrTestes for AG (green), VT (red), pluripotency-associated markers (cyan: POU5F1, NANOG), and DAPI (white), with their merges. Scale bars, 20 μm. **e** A dot plot showing the proportion of POU5F1+ or NANOG+ cells in hPGCLC-derived cells (TFAP2C+ or DDX4+) in d77 and d120 xrTestes as assessed by IF analysis on frozen sections. Each dot represents the proportion in one section, and the averages of 3 histologic sections are shown. The total number of POU5F1+ or NANOG+ cells in d77 (57 cells) or d120 xrTestes (11 cells) is also shown. Error bar, standard error of the mean (SEM). The statistical significance of the differences between d77 and d120 are evaluated by Fisher's exact test, $p = 0.002$. **f** Bright field (BF) images (top) and FACS analysis for AGVT expression (bottom) of xrTestes at d42, d81, and d124. Bar, 200 μm. **g** IF images of d120 xrTestes for AG (green), VT (red), T1 markers (cyan: MAGEA3, MAGEC2), and DAPI (white), with their merges. Scale bars, 20 μm. **h** FACS histogram of d81 and d124 xrTestes for PIWIL4-ECFP (PC) expression in the respective fraction of hPGCLC-derived cells. The percentage of PC+ cells in the respective fraction are shown. **i** IF images of hPGCLC-derived cells (TFAP2C or DDX4, green) in d77 or d120 xrTestes for MKI67 (red). Merges with DAPI (white) are shown on the right. Scale bars, 20 μm. **j** A dot plot showing the proportion of MKI67+ cells in hPGCLC-derived cells (TFAP2C+ or DDX4+) in d77 and d120 xrTestes. Each dot represents the proportion in one section and the averages of three histologic sections are shown. The total number of MKI67+ cells in d77 (85 cells) or d120 xrTestes (15 cells) is also shown. Error bar, SEM. The statistical significance of the differences between d77 and d120 are evaluated by Fisher's exact test, $p = 0.0047$. **k** IF analysis (left) and a box plot showing the fluorescence intensity of 5-methylcytosine (5mC, red) in hPGCLC-derived cells (green: DDX4) and other somatic cells counterstained with DAPI (white) in xrTestes at d120. Scale bar, 20 μm. hPGCLC-derived cells are outlined by white dotted lines. Center line, median; box limits, upper and lower quartiles; whiskers, 1.5× interquartile range. The statistical significance of the differences between GCs (Germ) and somatic cells (Soma) are evaluated by two-sided t-test assuming unequal variances. $p = 1.45 \times 10^{-5}$; $n$, the number of respective cells counted. See also Supplementary Figs. 2 and 3.

analysis (Fig. 3i, j). The expression of proapoptotic marker genes was downregulated or unchanged during T1LC specification, suggesting that T1LCs were quiescent but not overtly apoptotic (Fig. 4d).

Pairwise and multi-group DEG analyses among the six cell types we identified (hiPSCs, iMeLCs, hPGCLCs, MLCs, TCs, and T1LCs) showed the expression dynamics of genes involved in biologic processes that are unique to each cell type (Fig. 4e, Supplementary Figs. 6, 7, Supplementary Dataset 2). For example, iMeLCs exhibited a higher abundance of mRNAs encoding gene products involved in early mesodermal specification, such as *EOMES, MIXL1, CER1*, and genes related to SMAD signaling (e.g., *NODAL, LEFTY2*). During the iMeLC-to-hPGCLC transition, hPGCLCs began to express GC specifier genes and genes related to the GO terms "inflammation" and "apoptosis" (Supplementary Fig. 6 and Supplementary Dataset 2). The expression of genes involved in "piRNA pathways" and "spermatogenesis" became upregulated after the transition to MLCs (Supplementary Fig. 6).

The largest number of DEGs was upregulated in T1LCs (1,268 DEGs in a multi-group comparison); many of these DEGs were upregulated gradually during the transition from MLCs to T1LCs (Fig. 4e and Supplementary Fig. 7). In particular, we found genes involved in piRNA pathways and previously recognized markers for spermatogonia in DEGs for T1LCs (Fig. 4e, Supplementary Figs. 6, 7, Supplementary Dataset 3). Numerous transcription factors were also upregulated in T1LCs, such as *LMO4, ZNF451, TBP, NFXL1, EGR4, SCX*, and *EMX1*. Many of these were previously uncharacterized in the context of male germline development and may play important roles in the specification of prospermatogonia from PGCs (Fig. 4e, Supplementary Figs. 6, 7, Supplementary Dataset 3). GO terms enriched in T1LCs correspondingly included "transcription factor activity", "spermatogenesis", and "DNA methylation involved in gamete generation" (Fig. 4e). We also found that a significant fraction of genes known to be associated with human "spermatogenic failure" or "male infertility" were in T1LCs, suggesting the potential utility of our in vitro platform for identifying the molecular mechanisms of human male infertility (Fig. 4f).

**T1LCs exhibit a transcriptome similar to that of T1 in vivo.** After establishing our in vitro platform for human male GC

development, we compared our in vitro and in vivo transcriptome data to precisely define the status of our in vitro derived GCs along the developmental timeline. All three in vivo testicular samples were intermingled in a tSNE plot and segregated into clusters representing biologically meaningful cell types rather than sample identities (Supplementary Fig. 1c, g); however, a scatter plot comparing the averaged gene expression for either whole GCs or T1 across our three in vivo samples revealed a modest genome-wide reduction of mRNA abundance in Hs26 and Hs27 compared with Hs31 (Supplementary Fig. 1i). We believe this is a technical consequence of Hs26 and Hs27 being cryopreserved prior to analysis. Hs31 was prepared fresh and contained both M and T1 with differential expression patterns that matched the results obtained after aggregating all three samples (Supplementary Fig. 1h, j, k). To ensure the precision of gene expression comparisons with in vitro samples, which were all prepared fresh, we decided to use only Hs31 for downstream comparative studies.

We first performed unsupervised hierarchical clustering using averaged expression values for each cell type, which revealed two large clusters: a pre-germ cell/pre-gonadal phase cluster (hiPSCs, iMeLCs, hPGCLCs) and a prospermatogonial (gonadal) phase cluster (M, MLCs, TCs, T1, T1LCs) (Fig. 5a). Within the latter cluster, M/MLCs and T1/T1LCs each clustered together. Principal component analysis (PCA) on the prospermatogonial phase clusters revealed a gradual transition of cellular properties from M/MLCs to T1/T1LCs via TCs (Fig. 5b). Pairwise comparisons between cell types along this lineage trajectory demonstrated high concordance between in vivo and in vitro cell types; MLCs and T1LCs had the lowest number of DEGs and the highest coefficient of determination ($r^2$) when compared with M or T1, respectively (Fig. 5c). These results highlight the marked similarities between in vivo fetal male germline development and our in vitro platform. Nonetheless, we also noted modest differences in gene expression between T1 and T1LCs. The expression of some *HOXA* and *HOXB* family members was significantly higher in T1LCs (Fig. 5d and Supplementary Dataset 3), and reciprocally, *XIST*, a master regulator of X chromosome inactivation, that is expressed in human oogonia and prospermatogonia[34], was more highly expressed in T1, as were 18 other genes (Fig. 5d).

We additionally compared our results with published data from human male FGCs that were categorized into migrating, mitotic

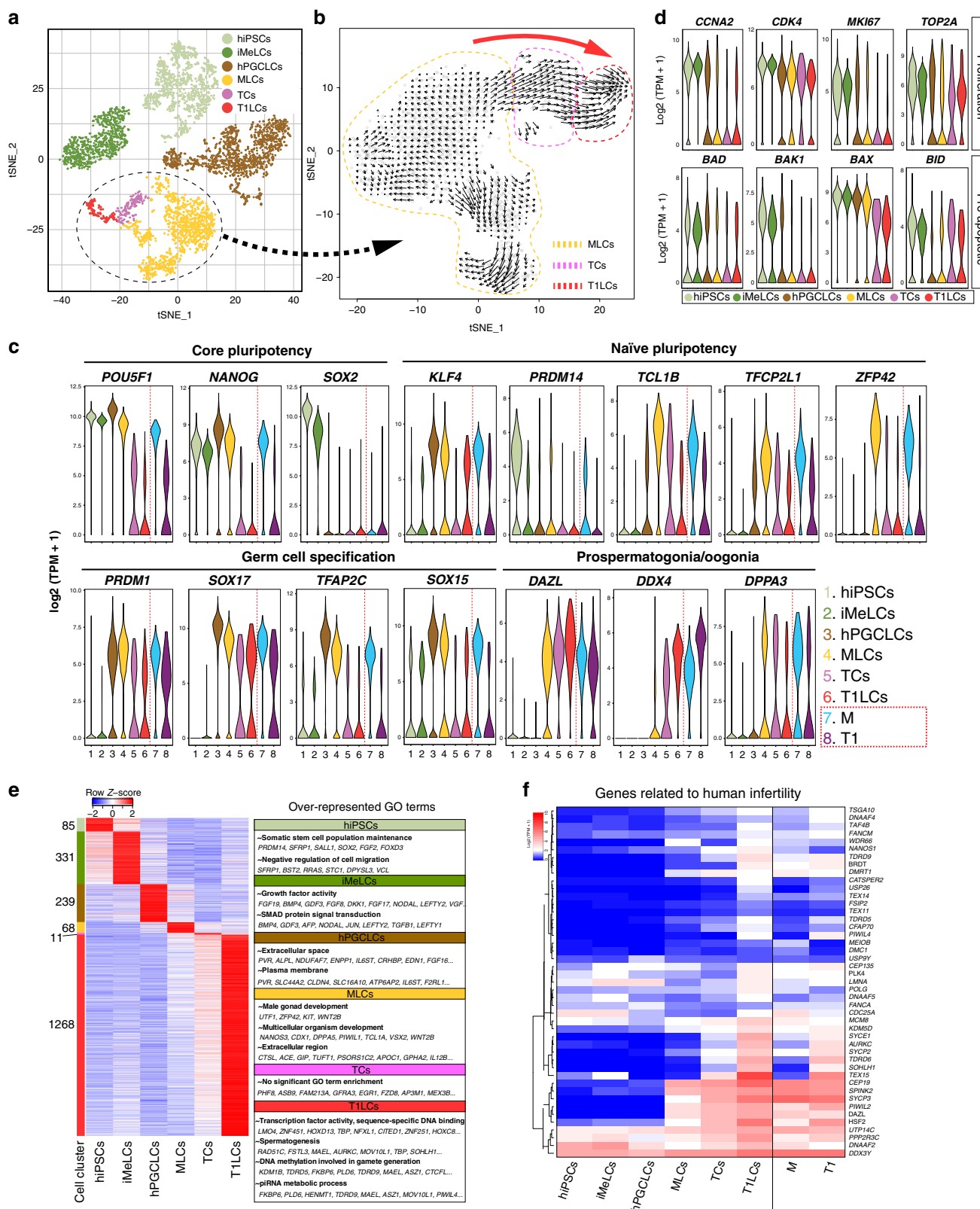

and mitotic-arrest FGCs[11]. In these datasets, migrating FGCs were collected from the aorta-gonad-mesonephros region of human male embryos at the age of 4 weeks. They were therefore categorized as PGCs in our study. The differentially expressed genes that distinguished these three cell types separated our hPGCLCs, M/MLCs, and T1/T1LCs/TCs largely into "migrating", "mitotic" and "mitotic-arrest" stages of human FGCs, respectively,

in agreement with direct comparison of our in vivo and in vitro specimens (Fig. 5e, Supplementary Dataset 4). We also performed PCA for all the cell clusters in our dataset alongside previously published data for neonatal human spermatogonia and found that T1LCs are slightly closer to neonatal spermatogonia than T1 (Supplementary Fig. 8a, b), which might partly explain the differences we observed between T1 and T1LCs (Fig. 5d).

**Fig. 4 Single-cell transcriptome profiling of in vitro derived human GCs. a** A tSNE plot of all *in vitro* cells using computationally aggregated scRNA-seq data obtained from six different samples (hiPSCs; iMeLCs; hPGCLCs_1; hPGCLCs_2, d81 xrTestis, d124 xrTestis). These cells are colored based on clusters defined in Supplementary Fig. 4b and c: hiPSCs (1168 cells, pale green); iMeLCs (800 cells, green), hPGCLCs (1371 cells, brown), MLCs (1224, yellow), TCs (162 cells, purple) and T1LCs (150 cells, red). The cluster re-analyzed in (**b**) is outlined by a dotted line. **b** Velocity analysis focused on a cluster outlined by a dotted line in (**a**) projected on the tSNE space defined in Supplementary Fig. 4f. Black arrows indicate the lineage trajectory estimated using nascent transcripts from scRNA-seq data in (**a**). Dotted lines enclose cells representing MLCs (yellow), TCs (purple) or T1LCs (red) as defined in Supplementary Fig. 4f. Large red arrow indicates the overall direction of the lineage trajectory at the borders of cell clusters. **c** Violin plots showing the expression of representative markers in in vitro (left) or in vivo (right) cell clusters defined in Figs. 1a, 4a. **d** Violin plots showing the expression of representative proliferation markers (top) or pro-apoptotic markers (bottom) in the respective cell clusters. **e** Heatmap showing the expression pattern of DEGs identified from a multi-group comparison (FDR < 0.01, fold change > 2 compared to other clusters) among cell clusters, and the over-represented GO terms in these DEGs. Color bars on the left indicate DEGs for respective cell clusters. The number of DEGs for each cell cluster are shown at the side of the color bar. **f** Heatmap of the expression of 45 genes in which mutations are associated with "spermatogenic failure" or "male infertility" in humans. Genes are ordered by UHC (Ward's method). See also Supplementary Figs. 4–7 and Supplementary Dataset 2.

**Comparison of T1LCs with oogonia-like cells induced from hiPSCs.** To highlight the similarities and differences between male and female GC development, we compared the transcriptome profiles of our in vitro T1LCs with published data for ag120 AG$^{+/-}$VT$^+$ oogonia-like cells induced from hiPSCs via intermediate states (i.e., iMeLCs, hPGCLCs and ag77 AG$^+$VT$^+$ cells)[15]. ag120 AG$^{+/-}$VT$^+$ and ag77 AG$^+$VT$^+$ cells in the previously published data roughly correspond to retinoic acid (RA)-responsive and mitotic in vivo female FGCs, respectively[11,15]. Among the 1177 DEGs that were unique to T1LCs and commonly annotated in both the male and female data sets, 316 DEGs were also similarly higher in oogonia-like cells (Fig. 5f and Supplementary Dataset 5), including many genes involved in piRNA biogenesis and de novo DNA methylation. We also identified 110 genes that were highly expressed in T1LCs but had weak or no expression in oogonia-like cells (Fig. 5g and Supplementary Dataset 5). These genes included many transcription factors, such as *HOXB4, HOXC9, TBX3, ESX1, SCX, SIX1*, and accordingly were enriched for GO terms such as "nucleus" and "sequence-specific DNA binding." Of note, our T1LCs were induced from hiPSCs bearing an XY karyotype. Accordingly, we found that 6 Y-chromosomal genes at the male-specific region of the Y chromosome (MSY) were differentially expressed among our in vitro dataset (Fig. 5h). Four of these genes (*DDX3Y, EIF1AY, KDM5D, TSPY2*) were significantly upregulated along the lineage trajectory. A similar trend was observed between M and T1.

**Expression dynamics of transposable elements (TEs) during human male germline development.** A hallmark feature of GC development is the activation and subsequent repression of TEs, which ensures the genome integrity of GCs while still permitting co-evolution between TEs and their hosts[16,35–37]. However, the precise expression dynamics of TEs in human male GC development is not well characterized. We used our in vitro platform, which covers at least the first ~20 weeks of human male GC development, to comprehensively map the expression patterns of various TE categories. Dimension reduction analysis of TE expression profiles obtained from scRNA-seq data demonstrated that each differentiation stage exhibits unique TE expression dynamics (Fig. 6a). Notably, M/MLCs and T1/T1LCs clustered together in both a UMAP plot and a hierarchical clustering analysis performed on differentially expressed TEs, further underscoring the similarities between in vivo cells and our in vitro model (Fig. 6a, b). Based on total TE-derived transcript abundances, TE were globally upregulated over time, both in vivo and in vitro, with the highest expression levels observed in T1 and T1LCs (Fig. 6c).

Most TE subtypes belonging to LINE1, SINE, DNA transposon, and SVA consistently showed gradual de-repression, presumably due to genome-wide DNA demethylation and other

epigenetic reprogramming events that accompany FGC development (Fig. 6d, e)[23,34,36]. In contrast to these TEs, LTR-type retrotransposons (i.e., endogenous retroviruses; ERVs) exhibited a more stage-specific expression pattern (Fig. 6d–f). For example, among ERV1 families, HERV-H and LTR7 expression peaked at hiPSCs and then subsequently declined (Fig. 6f), consistent with their roles in regulating the pluripotency gene network[38–40]. Another group of ERV1 families, LTR12 (LTR12C, LTR12D, LTR12_) were significantly upregulated in T1 or T1LCs (Fig. 6f). On the other hand, some members of the ERVK family, such as HERVIP10FH, were more highly expressed in hPGCLCs (Fig. 6f).

The stage-specific expression dynamics of ERVs suggests that they are involved in regulating the expression of genes involved in GC development. In fact, the analysis of publicly available ATAC-seq data revealed the enrichment of HERVH/LTR7 and HERVIP10FH in the ATAC-seq peaks obtained from ESCs/iMeLCs or hPGCLCs, respectively (Fig. 6g)[41,42], coinciding with their expression peaks in the respective cell types in our study (Fig. 6f). Moreover, LTR12 subfamilies were enriched in the ATAC-seq peaks from human male gonadal GCs and adult spermatogonia, suggesting that these TEs might be involved in gene regulation in human (pro)spermatogonia (Fig. 6g).

In summary, our observation that ERVs are regulated in a dynamic and stage-specific manner during male GC development raises the intriguing possibility that ERVs may be directly involved in the transcriptional activation of genes driving human prospermatogonial specification. Our observation additionally suggests that the identification of sequences downstream of these regulatory elements may reveal genes that play a causative role in this process (Fig. 6h).

## Discussion
Adult spermatogenesis is performed by SSCs that are capable of perpetual self-renewal. One potential treatment for SCOS in infertile males could therefore be deriving gametes from established human spermatogonial cultures either in vitro or, following transplantation, in vivo. In mice, cells with SSC activity can be propagated perpetually, which allows them to be characterized, genetically manipulated, and transplanted into infertile recipients to replenish spermatogenesis[43]. However, despite numerous attempts, sustained human spermatogonial cultures have yet to be convincingly established[43]. Moreover, defective gametogenesis in many infertile men occurs at a pre-meiotic stage, precluding the use of SSC-dependent reconstitution systems. This limitation could be overcome if male GCs could be derived from hiPSCs after in vitro correction of the mutations causing infertility. Several groups have recently derived functional male GCs from pluripotent stem cells in mice[30,44]. In this study, we provide the first evidence that hiPSCs can be used to reconstitute fetal human

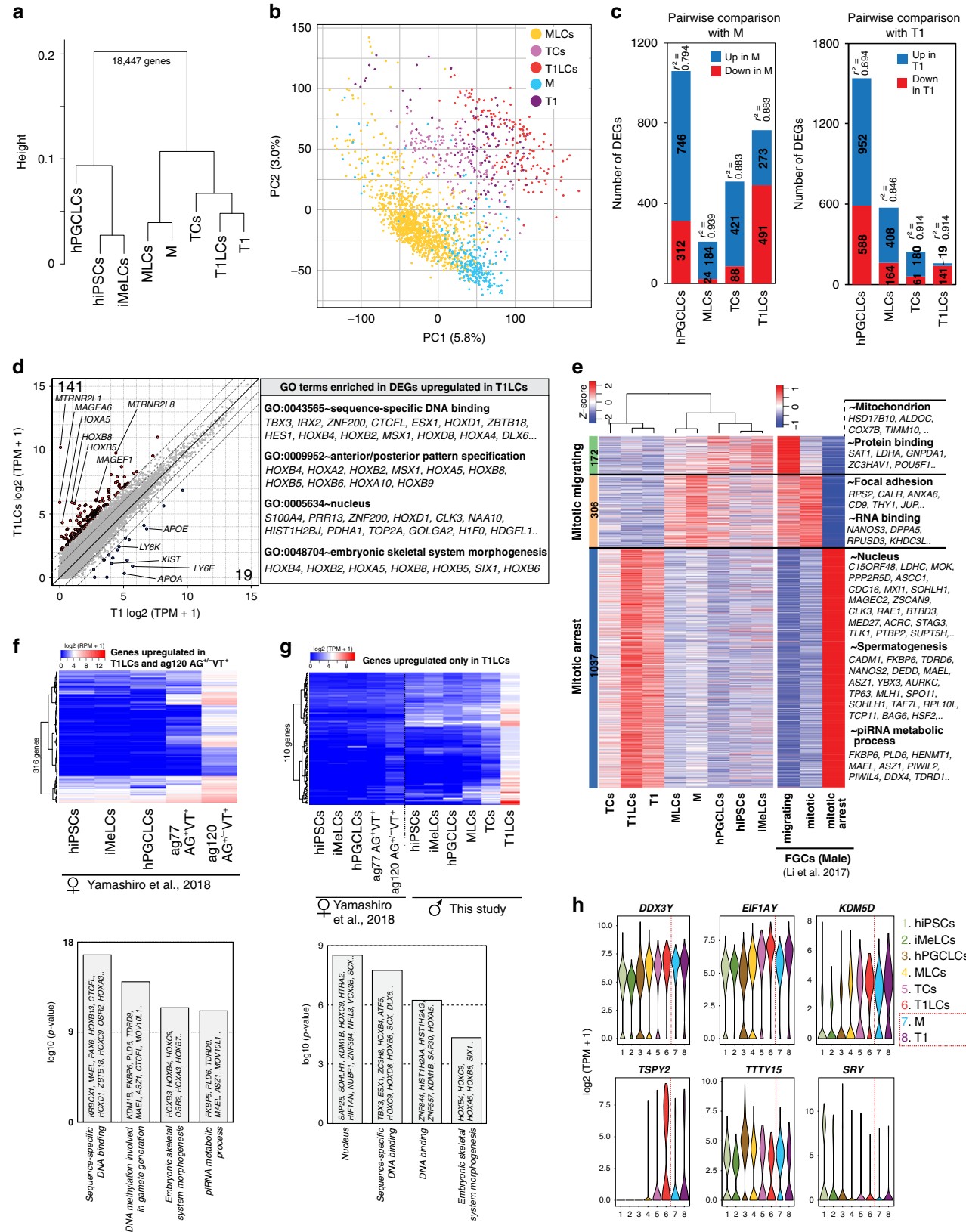

male germline development in vitro through stepwise and faithful recapitulation of the normal developmental processes (Fig. 6h).

We found that, in xrTestes, ~75% of hPGCLC-derived cells differentiated into VT⁺ MLCs or TCs by ~d80 of ALI culture (Fig. 3e). Although we did not further refine the exact timing at

which the hPGCLC-to-MLC transition occurs, the developmental kinetics of GCs in xrTestis appear to be somewhat faster than that of xrOvary, in which only a small fraction (~15%) of cells showed VT expression by ~d80 of ALI culture[15]. Moreover, the AG⁻VT⁺ cell fraction, which seems to be enriched for GCs at advanced

**Fig. 5 Comparison of lineage projection in vitro with that of in vivo. a** UHC of the averaged transcriptomes of in vivo (M, T1) and in vitro cell clusters (hiPSCs, iMeLCs, hPGCLCs, MLCs, TCs, T1LCs) defined in Figs. 1a, 4a. **b** PCA for prospermatogonial stage GCs (M, MLCs, TCs, T1, and T1LCs). Color codes for cell clusters are indicated. **c** Number of DEGs and coefficient of determination ($r^2$) for pairwise comparisons between in vivo (M [left] or T1 [right]) and in vitro clusters (hPGCLCs, MLCs, TCs, and T1LCs). DEGs are defined as genes with more than fourfold differences between two groups (mean log2[TPM +1] > 2, FDR < 0.01). $r^2$ are calculated using all annotated genes (18447 genes). **d** Scatter plot comparison of the averaged values of gene expression between T1 and T1LCs. Blue, genes higher in T1; red, genes higher in T1LCs (more than 4-fold differences [flanking diagonal lines], mean log2[TPM+1] > 2, FDR < 0.01). Key genes are annotated and the number of DEGs are indicated. Representative genes and their GO enrichments for genes higher in T1LCs are shown on the right. **e** Heatmaps of the expression of markers for "migrating" (172 genes), "mitotic" (306 genes), and "mitotic-arrest" (1037 genes) human male FGCs (defined by Li et al.[11]) in the respective cell clusters defined in this study (left) and the indicated male FGC types defined by Li et al.[11] (right). Using these markers, hierarchical clustering was performed for cell clusters defined in this study (left). GO terms for the respective markers are shown on the right. **f** Heatmap of the averaged expression values of 316 genes in indicated cell types defined by Yamashiro et al.[15] (top). Among markers for T1LCs defined in Fig. 4e, 1177 genes were also annotated in the dataset by Yamashiro et al.[15]. Among them, 316 genes upregulated in ag120 AG$^{+/-}$VT$^+$ oogonia-like cells relative to all other cell types (more than twofold difference) are shown (top). GO analysis for these 316 genes are also shown (bottom). **g** Heatmap of the averaged expression in the indicated cell types for 110 genes highly expressed in T1LCs but with weak or no expression in ag120 AG$^{+/-}$VT$^+$ oogonia-like cells. To enable comparison between two different scRNA-seq platforms, RPM values of the data from Yamashiro et al. were adjusted using a polynomial regression curve (see "Methods"). Among 1,177 DEGs for T1LCs (Fig. 4e), 110 genes showing high levels of expression in T1LCs (mean log$_2$[TPM+1] > 4) and low levels of expression in oogonia-like cells (adjusted log$_2$[RPM+1] < 2) are shown. GO analysis for these 110 genes are also shown (bottom). **h** Violin plot showing the expression levels of genes at the male-specific regions of the Y chromosome (MSY) in indicated cell clusters. These genes were identified the by multi-group DEG analysis in Fig. 4e (without cut-off by fold-change > 2). See also Supplementary Figs. 4, 6, 7, Supplementary Dataset 3–5.

---

stages (TCs, T1LCs or RA-responsive oogonia-like cells), contained ~13% of GCs in xrTestis at ~d80 compared to only 3% of GCs in xrOvaries at ~d100, suggesting that xrTestes might provide a better niche than xrOvaries for promoting sex-specific GC progression into advanced stages. Future studies directly comparing the two culture platforms using the same hiPSC lines may be warranted to determine if this finding is universal.

Our xrTestis culture revealed the protracted time course for the progression from hPGCLCs to T1LCs (~120d), particularly given that the progression from PGCs to T1 takes only 1 week in mice[8]. According to a previous study[11], T1 emerges as early as 9 weeks post fertilization. Therefore, the emergence of T1LCs is slower than the fastest timeline of T1 development in vivo. However, given that T1 and M were simultaneously present in all our testicular samples at 17 to 18 weeks and in samples from a previous study up to 26 weeks[11], some GCs may not progress into T1 until the third trimester. In this regard, the developmental timeline of our T1LCs may still be within a physiological range.

In extended cultures, we noted that the majority of tubular structures dissolved after ~d80, and as a result, GCs were randomly localized in the stroma and lost contact with SCs by d120. The fate of T1LCs appeared to be maintained until at least d120, even in the absence of tubular structure; however, the absence of a tubular structure after d120 will likely pose significant challenges for further differentiation of T1LCs into more advanced male GC types. In particular, it may be difficult to derive neonatal or adult spermatogonia, which require continuous interaction with the environment provided by SCs[17–19]. The underlying reasons for in vitro tubular dissolution are still unclear, but one possibility is that long-term cultured SC may more closely reflect their postnatal counterparts and therefore have different nutritional requirements. Fetal Leydig cells are known to be essential for Sertoli cell proliferation and tubulogenesis in mice through the production of Activin A, therefore loss of fetal Leydig cells in xrTestes might also contribute to the tubular dissolution after long-term culture[45]. Another potential reason is the lack of endothelial cells in xrTestes, which was evident as early as d14 (Fig. 2), because the vasculature has a well-established role in testicular cord formation and proper structural testis development[46,47]. Supplementing xrTestes with cultured testicular endothelial cells or growth factors that sustain their survival might help vascularize xrTestes. To establish more robust, sustainable, and complete in vitro reconstitution of fetal

spermatogenesis, future studies may also seek to replace the xrTestis constituents with more physiological niches, such as those derived from the human fetal testis or their mimetics induced from hiPSCs.

Despite its limitations, our xrTestis culture approach is the first method that reliably produces T1-stage human fetal prospermatogonia from hiPSCs in vitro. Surprisingly, we found that our T1LCs express a number of known marker genes for mouse or human spermatogonia, such as *SIX1, DCAF4L1, PLPPR3, EGR4*, and *ID4*, among others (Figs. 4, 5, Supplementary Figs. 6, 7, Supplementary Dataset 2)[17,18,28,48]. Moreover, our comparison of the T1LCs and T1 transcriptomes with previous data for neonatal prospermatogonia suggested that T1LCs appear to be closer to neonatal prospermatogonia than T1 (Supplementary Fig. 8). Due to the paucity of available data on perinatal human male germline development, the exact in vivo stage to which T1LCs correspond remains unclear. Recent studies have suggested that SSC activity, or the competency to produce self-sustaining spermatogenesis upon transplantation, has already been established, at least in part, in fetal prospermatogonia in mice[49]. It is therefore possible that T1LCs might be close to foundational SSCs, which upon transplantation, would be capable of regenerating complete spermatogenesis. Experimental validation of this possibility would require a relevant model, such as non-human primates, in which the full regenerative potential of T1LCs could be assessed[50,51].

T1LCs derived from our xrTestis culture also expressed many genes linked to human male infertility, raising the possibility of using the xrTestis platform to dissect the cellular and molecular mechanisms of human male infertility (Figs. 4, 5). For example, among the DEGs we identified, *DDX3Y* has been suggested to play a critical role in fetal male GC development and the mutation appears to be responsible for one of the common causes of SCOS caused by Y-chromosome microdeletions[52].

One suite of DEGs identified in this study (*DDX3Y, EIF1AY*, and *KDM5D*) are Y-chromosomal genes at the MSY. Expression of these genes were detectable at M and earlier stages, which inevitably confer sexual dimorphism between male and female GCs at least at the molecular level. Indeed, male and female mitotic FGCs, which are present at the earliest stage of the gonad (4–5 weeks post fertilization), already exhibit differential expression of many sex-chromosomal and autosomal genes[11]. Further investigation will be needed to determine when and how

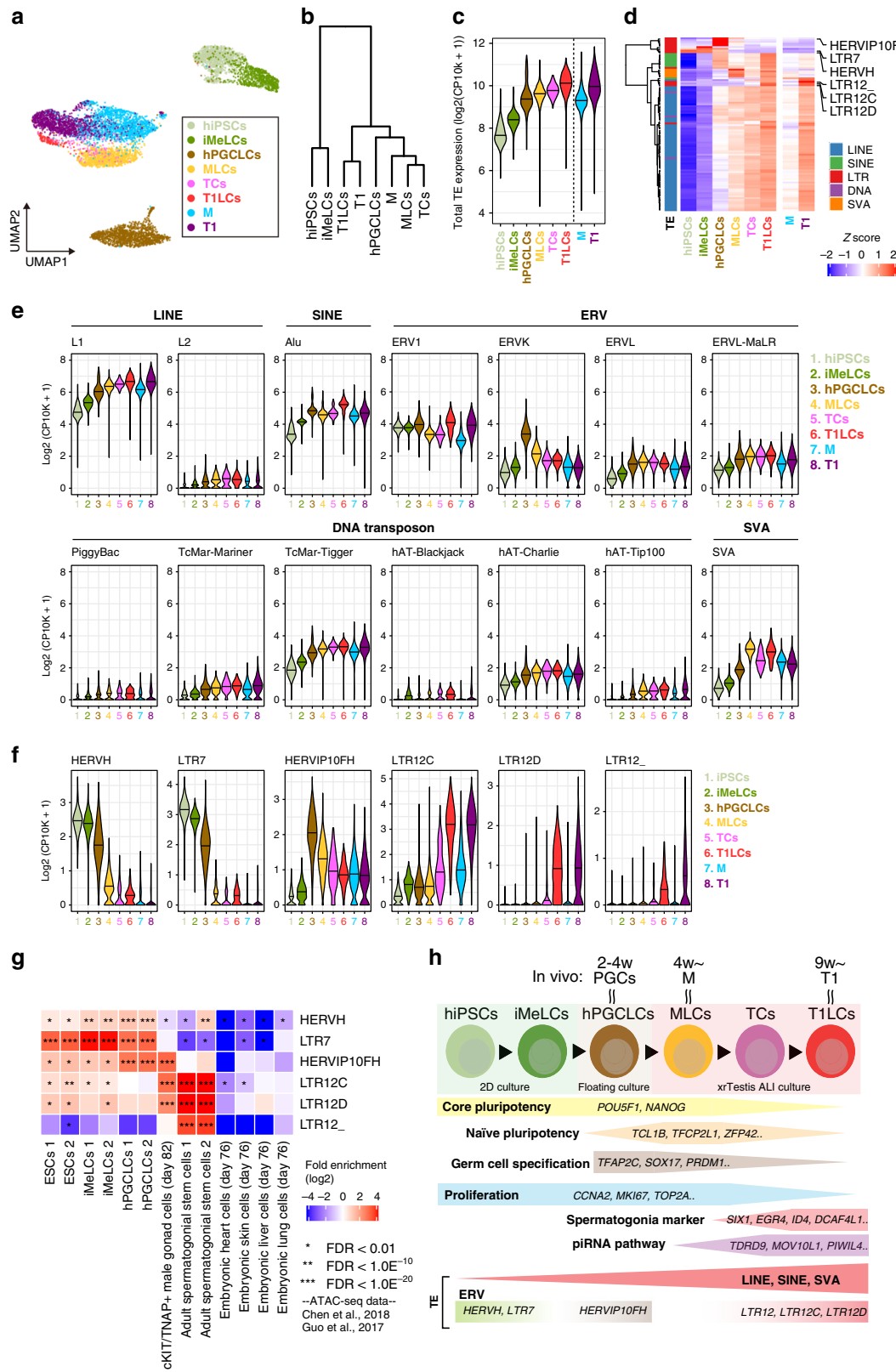

sexual dimorphism arises in human GCs; our xrTestis platform, along with the previously established xrOvary system, can provide the basis for evaluating sexual determination in the human germline in a controlled setting.

Our transcriptome analysis additionally revealed that T1LCs expressed genes involved in piRNA biogenesis (Figs. 4, 5, Supplementary Figs. 6, 7). This observation suggests that the xrTestis

platform could be used to address the molecular mechanisms of piRNA production and the consequent regulation of various TEs, which are fossilized viral descendants that can have detrimental effects on genome integrity if they are not properly controlled during GC development[35]. Recent studies have highlighted the key role of LTRs as *cis*-regulatory elements in the transcriptional regulation of genes involved in early embryogenesis. These

**Fig. 6 Dynamic regulation of TE expression during human male germline development. a** Dimension reduction analysis of scRNA-Seq data based on TE expression profile. The 100 most variably expressed TE subfamilies were used in the analysis. **b** Hierarchical clustering (Ward's method using Euclidian distances) of the respective cell clusters (averaged expression values) using the 200 most variably expressed TEs. **c** Dynamics of the total expression level of TEs. **d** Heatmap showing the expression dynamics of respective TE subfamilies. The 200 most variably expressed TEs are shown. Classifications of TEs are shown in the left of heatmap. **e** Expression dynamics of respective TE families. **f** Expression dynamics of ERV subfamilies. ERVs exhibiting unique expression patterns are shown. **g** Enrichment of ATAC-Seq signals on specific ERV subfamilies. The fold-enrichment of the overlaps between ATAC-Seq peaks and a specific ERV subfamily was calculated on the random expectation. Publicly available ATAC-Seq data (Chen et al.[41] and Guo et al.[42]) were used. **h** A model for T1LC induction from hiPSCs using xrTestis culture.

studies hypothesized that that host-viral coevolution exploited TEs that are typically upregulated during early embryo or germline development due to the epigenetically permissive genomic status[53–55]. Consistent with this hypothesis, we found that LTR12 subfamilies were uniquely expressed among T1LCs and T1s and were positionally enriched in ATAC-seq peaks identified in human prospermatogonia or spermatogonia. These LTRs may serve as cis-regulatory elements for germline development (Fig. 6). Future studies using CRISPRa or CRISPRi[53] to alter the expression of LTRs in xrTestes might enable a functional interrogation of the role of LTRs in human GC development.

In conclusion, we demonstrate that human fetal male GC development can be reconstituted from hiPSCs using our in vitro xrTestis platform (Fig. 6h). We believe this new platform provides a useful resource for studying various aspects of human male gametogenesis and a significant step forward in the development of in vitro human sperm production.

## Methods

**Mice.** Animal procedures were conducted in compliance with the Institutional Animal Care and Use Committee of the University of Pennsylvania (IACUC Protocol #806671). Timed pregnant ICR female mice were purchased from Charles River Laboratories (Wilmington, MD). Mice were housed under a 12:12 h light: dark cycle at 22–24 °C, with a humidity of 40–60%.

**Collection of human fetal samples.** Samples of fetal testes (sample codes: Hs26, Hs27, Hs31) were obtained after 17–18 weeks of gestation from three donors undergoing elective abortion at the University of Pennsylvania hospital. All experimental procedures were approved by the Institutional Review Board (IRB; Protocol #832470) and Human Stem Cell Research Advisory Committee at the University of Pennsylvania. Informed consent was obtained from all human subjects.

**Culture of hiPSCs.** All experiments inducing human primordial germ cell-like cells (hPGCLCs) from hiPSCs were approved by the University of Pennsylvania IRB. hiPSCs were cultured on plates coated with Recombinant laminin-511 E8 (iMatrix-511 Silk, Nacalai USA) and were maintained under a feeder-free condition in the StemFit® Basic04 medium (Ajinomoto) containing basic FGF (Peprotech) at 37 °C under an atmosphere of 5% $CO_2$ in air. Prior to passaging or the induction of differentiation, hiPSC cultures were treated with a 1:1 mixture of TrypLE Select (Life Technologies) and 0.5 mM EDTA/PBS for 14 min at 37 °C to dissociate them into single cells. 10 μM ROCK inhibitor (Y-27632; Tocris) was added in culture medium for 1 day after passaging hiPSCs.

**Human testis sample preparation.** The sex of each sampled fetus was determined by sex-specific PCR. PCR primers targeted the ZFX/ZFY loci (Supplementary Dataset 6)[56], and genomic DNA isolated from the mesonephros and attached fibroconnective tissues was used as the template. Fetal testes were dissected out in RPMI-1640 (Roche) and ~one-fourth of the tissues were fixed in 10% formalin overnight at room temperature before processing for histologic analyses.

The remaining testicular tissue was dissociated into single cells using a two-step enzymatic digestion protocol. In brief, fetal gonads were cut into small pieces (~1 mm²) and digested with type IV collagenase for 8 min at 37 °C. Tissue pieces were centrifuged at $200 \times g$ for 5 min, washed with Hank's Balanced Salt Solution (Thermo Fisher Scientific), and then digested with 0.25% trypsin/EDTA and DNase I for 8 min at 37 °C. The digestion was quenched by adding 10% of fetal bovine serum. Cells were then strained through a nylon cell strainer (70 μm pore size) and pelleted by centrifugation at $200 \times g$ for 5 min Hs26 and Hs27 were subsequently resuspended in Cell Banker Type I (Amsbio, ZNQ CB011) and cryopreserved in liquid nitrogen until use. Hs31 was processed without cryopreservation.

Before use, cryopreserved cells were thawed at 37 °C and washed once with 5 ml of MACS buffer (PBS containing 0.5% BSA and 2 mM EDTA). Dead cells were subsequently removed by dead cell removal microbeads (Miltenyi Biotec). Flow-through cells were pelleted by centrifugation at $600 \times g$ for 15 min, resuspended in PBS containing 0.1% BSA, and then loaded into the 10× Genomics Chromium Controller.

**Generation of AGVTPC-knock-in reporter lines.** The donor vector and the TALEN constructs for generating the *DDX4/hVH-p2A-tdTomato* (VT) allele were described previously[15]. In brief, homology arms of *DDX4* (left arm:1471 bp; right arm: 1291 bp) were PCR-amplified from the genomic DNA of *TFAP2C-p2A-EGFP* (AG) male hiPSCs (585B1 1-7, gift from Drs. Kotaro Sasaki and Mitinori Saitou, Kyoto University) and was sub-cloned into the pCR2.1 vector using the TOPO TA cloning kit (Life Technologies). The *p2A-tdTomato* fragment with the PGK-Neo cassette flanked by *loxP* sites was also PCR-amplified and inserted in-frame at the 3′-end of the *DDX4* coding region. TALEN constructs targeting *DDX4* were generated using a Golden Gate TALEN and TAL Effector Kit 2.0 (Addgene, #1000000024)[57]. TALEN's RVD sequences were as follows: DDX4-left (5-prime), HD HD HD NI NI NG HD HD NI NN NG NI NN NI NG NN NI NG NN; DDX4-right (3-prime), NN NI NI NN NN NI NG NN NG NG NG NG NN NN HD NG NG[15].

The homology arms of *PIWIL4* (left arm: 1497 bp; right arm: 1000 bp) were similarly PCR-amplified and sub-cloned to construct the donor vector for generating the *PIWIL4-p2A-ECFP* (PC) allele. The *p2A-ECFP* fragment with the PGK-Puro cassette flanked by *loxP* sites was synthesized by GeneUniversal (Newark, DE) and inserted in-frame at the 3′-ends of *PIWIL4* coding sequence using the GeneArt Seamless Cloning & Assembly Kit (Life Technologies). The *PIWIL4* stop codon was removed to allow for expression of the in-frame *p2A-ECFP* protein. A *MC1-DT-A-polyA* cassette was synthesized and subsequently inserted into the downstream region of the right homology arm of the PC donor vector using the restriction enzymes NotI/XhoI.

The pair of single-guide RNAs (sgRNAs) targeting the sequence close to the *PIWIL4* stop codon was designed by the Molecular Biology CRISPR design tool (Benchling). Two sgRNAs (GAACTACTGGCATCACTAGA and TCACAGGTAGAAGAGATGGT) were selected and cloned into the pX335-U6-Chimeric BB-CBh-hSpCas9n (D10A) SpCas9n-expressing vector to generate the sgRNAs/Cas9n vector (Addgene, #42335)[58]. The recombination activity of the sgRNAs/Cas9n vector was validated using a single-strand annealing (SSA) assay.

The donor vectors (5 μg) and TALEN and sgRNAs/nCas9 vectors (2.5 μg each) for *DDX4* and *PIWIL4* were introduced into one million AG hiPSCs (585B1 1-7, male) by electroporation using a NEPA21 Type II Electroporator (Nepagene). Single colonies were isolated after selection with puromycin and neomycin and subsequent transfection with a plasmid expressing Cre recombinase to remove the *PGK-Puro* and *PGK-Neo* cassettes. The success of the targeting, random integration and Cre recombination process was assessed by performing PCR on extracted genomic DNA from each colony using the primer pairs listed in Supplementary Dataset 6.

**Karyotyping and G-band analyses.** hiPSCs were incubated in culture medium containing 100 ng/ml of KaryoMAX Colcemid solution (Gibco) for 8 h. After dissociation by TrypLE Select, cells were treated with pre-warmed Buffered Hypotonic solution and incubated for 30 min at 37 °C. Cells were then fixed with Carnoy's solution (3:1 mixture of methanol and acetic acid) and dropped onto glass slides to prepare the chromosomal spread. Karyotypes were first screened by counting the numbers of chromosomes identified by DAPI staining. Cell lines bearing 46 chromosomes were further analyzed by G-banding performed by Cell Line Genetics (Madison, WI).

**Induction of hPGCLCs and the generation of xrTestes.** hPGCLCs were induced from hiPSCs via iMeLCs as described previously[14]. For the induction of iMeLCs, hiPSCs were plated at a density of $4–5 \times 10^4$ cells/cm² onto a human fibronectin (Millipore)-coated 12-well plate in GK15 medium (GMEM [Life Technologies]) with 15% KSR, 0.1 mM NEAA, 2 mM L-glutamine, 1 mM sodium pyruvate and 0.1 mM 2-mercaptoethanol) containing 50 ng/ml of ACTA (R&D Systems, 338-AC), 3 μM CHIR99021 (Tocris Bioscience, 4423) and 10 μM Y-27632 (Tocris, 1254). After 31–38 h, iMeLCs were harvested and dissociated into single

cells with TrypLE Select. Cells were then plated into a low-cell-binding V-bottom 96-well plate (Greiner) at 3500 cells per well for induction into hPGCLCs. Wells contained GK15 medium supplemented with 200 ng/ml BMP4 (R&D Systems, 314-BP-010), 100 ng/ml SCF (R&D Systems, 255-SC-010), 50 ng/mL EGF (R&D Systems, 236-EG), 1000 U/ml LIF (Millipore, #LIF1005) and 10 μM of Y-27632.

xrTestes were generated by aggregating d5 hPGCLCs with fetal testicular somatic cells obtained from E12.5 mouse embryos following a previously described procedure[30]. d5 hPGCLCs were obtained using fluorescence-activated cell sorting (FACS; full procedure described later in this section). To isolate fetal testicular somatic cells, E12.5 embryos were isolated from timed pregnant ICR females and collected in chilled DMEM (Gibco) containing 10% FBS (Gibco) and 100 U/ml penicillin/streptomycin (Gibco). Fetal male testes were identified by their appearance, and the mesonephros were removed by tungsten needles.

Isolated E12.5 testes were washed with PBS and then incubated with dissociation buffer for 15 min at 37 °C with periodic pipetting. The dissociation buffer contained 1 mg/ml Hyaluronidase Type IV (Sigma), 5 U Dispase (Corning), and 5 U DNase (Qiagen) in wash buffer (100 U/ml penicillin/streptomycin and 0.1% BSA in DMEM/F12). After another PBS wash, testes were dissociated into single cells using 0.05% Trypsin-EDTA in PBS for 10 min at 37 °C followed by quenching with FBS. Cell suspensions were strained through a 70 μm nylon cell strainer and centrifuged. The remaining cell pellet was then resuspended with MACS buffer (PBS containing 0.5% BSA and 2 mM EDTA) and incubated with anti-SSEA1 antibody MicroBeads (Miltenyi Biotec) for 20 min on ice before being washed with MACS buffer and centrifuged. The cell pellet was again resuspended in MACS buffer and then applied to an MS column (Miltenyi Biotec) according to the manufacturer's protocol. The flow-through cells were centrifuged, resuspended with Cell Banker Type I, and cryopreserved in liquid nitrogen until use. All centrifugations were performed at $232 \times g$ for 5 min and were followed by removal of the supernatant.

To generate floating aggregates, hPGCLCs (5000 cells per xrTestis) and thawed fetal testicular somatic cells (60,000 cells per xrTestis) were mixed and plated in a Lipidure-coated U-bottom 96-well plate (Thermo Fisher Scientific, 174925) in Minimum Essential Medium alpha (α-MEM, Invitrogen) containing 10% KSR (Gibco), 55 μM 2-mercaptoethanol (Gibco), 100 U/ml penicillin/streptomycin (Gibco) and 10 μM Y-27632. After two days of floating culture, floating aggregates were transferred onto Transwell-COL membrane inserts (Corning, 3496) using a glass capillary. Membrane inserts were soaked in α-MEM supplemented as before, without Y-27632. xrTestes were cultured at 37 °C under an atmosphere of 5% $CO_2$ in air and one-half the volume of medium was changed every three days.

**IF analysis**. For IF analysis of xrTestes tissue, xrTestes were fixed with 2% paraformaldehyde (Sigma) in PBS for 3 h on ice, washed three time with PBS containing 0.2% Tween-20 (PBST), and then successively immersed in 10% and 30% sucrose (Fisher Scientific) in PBS overnight at 4 °C. The fixed tissues were embedded in OCT compound (Fisher Scientific), frozen, and sectioned to 10 μm thickness using a −20 °C cryostat (Leica, CM1800). Sections were placed on Superfrost Microscope glass slides (Thermo Fisher Scientific) that were then air-dried and stored at −80 °C until use.

Prior to staining, slides were washed three times with PBS and then incubated with blocking solution (5% normal goat serum in PBST) for 1 h. Slides were subsequently incubated with (1) primary antibodies (Supplementary Dataset 7) in blocking solution for 2 h, followed by (2) secondary antibodies and 1 μg/ml DAPI in blocking solution for 50 min Both incubations were performed at room temperature and followed by four PBS washes. Slides were mounted in Vectashield mounting medium (Vector Laboratories) for confocal laser scanning microscopy analysis (Leica, SP5-FLIM inverted). Confocal images were processed using Leica LasX (version 3.7.2).

For IF analyses of fetal testes and some of the xrTestes tissue, samples were fixed in 10% buffered formalin (Fisher Healthcare) with gentle rocking overnight at room temperature. After dehydration, tissues were embedded in paraffin, serially sectioned at 4 μm thickness using a microtome (Thermo Scientific Microm™ HM325), and placed on Superfrost Microscope glass slides. Paraffin sections were then de-paraffinized using xylene. Antigens were retrieved by treating sections with HistoVT One (Nacalai USA) for 35 min at 90 °C and then for 15 min at room temperature. The staining and incubation procedure for paraffin sections was similar to that for frozen sections, with the following modifications: the blocking solution was 5% normal donkey serum in PBST; the primary antibody incubation was performed overnight at 4 °C; and slides were washed with PBS six times after each incubation. Slides were mounted in Vectashield mounting medium for confocal microscopic analysis.

IF targeting 5mC was performed using paraffin sections, as described above, with minor modifications. After being stained with primary and secondary antibodies, but not the anti-5mC antibody, slides were treated with 4N HCl in 0.1% Triton X for 10 min at room temperature followed by two brief washes with PBS, one 15 min wash with PBST, and another blocking step. Slides were then incubated with primary antibodies (anti-5mC and other antibodies) followed by secondary antibodies.

**Combined IF and ISH**. ISH on formalin-fixed paraffin-embedded sections was performed using the ViewRNA ISH Tissue Assay Kit (Thermo Fisher Scientific) with gene-specific probe sets for human *PIWIL4* (VA1-3014459VT), human *ACTB*

(VA1-103510VT, positive control) and *Bacillus subtilis dapB* (VF1-11712VT, negative control). Experiments were performed according to the manufacturer's instructions. After briefly washing once with PBS, these slides were used for IF studies as described above except without the antigen retrieval process. All human testicular tissues were confirmed to be ubiquitously reactive for *ACTB* and negative for *dapB* by confocal imaging.

**Fluorescence-activated cell sorting (FACS)**. We obtained the d5 hPGCLCs used for xrTestis formation using FACS. d5 floating aggregates containing hPGCLCs, which were induced from AGVTPC hiPSCs, were dissociated into single cells with 0.1% Trypsin/EDTA treatment for 15 min at 37 °C with periodic pipetting. After the reaction was quenched by adding an equal volume of FBS, cells were resuspended in FACS buffer (0.1% BSA in PBS) and strained through a 70 μm nylon cell strainer (Thermo Fisher Scientific) to remove cell clumps. $AG^+$ cells were sorted by FACSAria Fusion (BD Biosciences) and collected in an Eppendorf tube containing α-MEM. xrTestes-derived GCs were analyzed and sorted using a similar procedure, except 0.1% BSA fraction V in PBS was used to wash the cells prior to straining and $AG^{+/-}VT^{+/-}PC^{+/-}$ cells were collected in CELLOTION (Amsbio). All FACS data were collected using FACSDiva Software v 8.0.2 (BD Biosciences).

**10x Genomics single-cell RNA-seq library preparation**. FACS-sorted in vitro cells (hPGCLCs and d81 and d124 xrTestis samples) were collected in CELLO-TION. hiPSCs and iMeLCs were collected in the StemFit Basic 04 and GK15 media, respectively, without FACS sorting. Cells were centrifuged at $300 \times g$ for 5 min and then resuspended in 0.1% BSA in PBS.

Cells were loaded into chromium microfluidic chips with the Chromium Single Cell 3' Reagent Kit (v3 chemistry) and then used to generate single-cell gel bead emulsions (GEMs) using the Chromium Controller (10× Genomics) according to the manufacturer's protocol. GEM-RT was performed in a C1000 Touch Thermal Cycler with 96-Deep Well Reaction Module (Bio-Rad). All subsequent cDNA amplification and library construction steps were performed according to the manufacturer's protocol. Libraries were sequenced using a 2 × 150 paired-end sequencing protocol on an Illumina HiSeq 4000 or NovaSeq 6000 instrument.

### Quantification and statistical analysis

*Mapping reads of 10x chromium scRNA-seq and data analysis.* Raw RNA-seq reads were de-multiplexed using the mkfastq command in Cell Ranger (v3.1.0) and trimmed (28 bp cell barcode and UMI Read1, 8bp i7 index, and 91bp Read2) using cutadapt (v2.6). Trimmed sequence files were mapped to the human reference genome (GRCh38-3.0.0). Read counts were obtained from outputs made by Cell Ranger (v3.1.0) using edgeR (v3.22.3). Cells with fewer than 1800 genes and genes with fewer than 10 reads were removed, as were genes detected in only five or fewer cells. Secondary data analyses were performed using Microsoft Excel and R software version 3.6.1 with the packages ggplot2 (v3.2.1), gplots (v3.0.1.1), q value (v2.18.0), sp (v1.4-1), and TCC (v1.12.1). All analyses of expression data were performed using log2(TPM+1) values. Unsupervised hierarchical clustering (UHC) was performed using the hclust function with Euclidean distances and either Ward's (ward.D2) or the complete (complete) agglomeration method. PCA was performed using the prcomp function without scaling. tSNE analysis was performed using the Rtsne function and visualized using ggplot2. RNA velocity analysis was performed using Velocyte.R (v0.6)[26] (http://velocyto.org/) on cells filtered using the same selection criteria described above.

To identify genes that were differentially expressed among cell types (i.e., marker genes), we used the edgeR function from the TCC package (v1.20.1) to isolate genes with an FDR < 0.05 based on generalized linear model approach[59]. The DEGs were then defined as genes exhibiting >2-fold higher expression in one cluster compared to the remaining clusters. DEGs for each pairwise comparison among cell types were defined as genes with a minimum mean abundance of log2 (TPM+1) > 2, fold-change ≥4, and an FDR < 0.01 following Welch's *t*-test with the Benjamini-Hochberg correction. These DEGs were plotted over the scatter plot of averaged transcriptome values for cell clusters. DEGs were mapped to GO terms using DAVID (v. 6.8)[60] with the background list set to "Homo sapiens". Only enriched GO terms with *p* < 0.05 were shown.

To evaluate the expression of human infertility genes in our cell clusters, we retrieved 105 genes from Cannarella et al.[61] and the Online Inheritance in Man (OMIM) database using search terms of "male infertility" and "spermatogenic failure." Genes associated with pituitary, hypothalamus and gonadal somatic cells were excluded. Among these 105 genes, 45 genes with an average expression value of log2(TPM+1) > 1 in at least one cell cluster were selected for analysis.

We compared our results to a study of oogonia-like cells derived from female hiPSCs[15] by obtaining the gene expression matrix file (log2[RPM + 1] values) from that study. Given the timing of the culture and AG/VT expression status, we considered that female hiPSCs, iMeLCs, hPGCLCs, ag77AG+VT+, and Ag120 $AG^{+/-}VT^+$/Ag120AG−VT+ roughly correspond to our male hiPSCs, iMeLCs, hPGCLCs, MLCs, and TCs/T1LCs, respectively. We averaged the expression values for each cell type in the female dataset; Ag120 $AG^{+/-}VT^+$ and Ag120AG−VT+ samples were not replicated, so we used the average of these two samples for Ag120 $AG^{+/-}VT^+$ oogonia-like cells. The ag77AG+VT+ sample was also not replicated and was therefore used as is. Genes commonly upregulated in T1LCs in this study

and in ag120 AG$^{+/-}$VT$^+$ oogonia-like cells from Yamashiro et. al. were identified using the following criteria. First, we considered only T1LC marker genes, as defined by the DEG analysis described above (expression 2-fold higher than any other cell cluster in this study, FDR < 0.01), that were also annotated in the dataset by Yamashiro et al. Among these 1177 genes, 316 showed >2-fold higher expression in ag120 AG$^{+/-}$VT$^+$ oogonia-like cells compared to all other cell types in the dataset. The expression levels of these genes were shown in the heatmap.

Genes highly expressed in T1LCs but with weak or no expression in ag120 AG$^{+/-}$VT$^+$ oogonia-like cells were selected using the follow criteria. First, to enable comparison between two different scRNA-seq platforms, RPM values from Yamashiro et al. were adjusted using the polynomial regression curve $y = 0.0908x^2 - 0.0454x + 0.1243$ (least square method), which we defined from a scatter plot comparison of hiPSCs from this study and hiPSCs from Yamashiro et al. Among the 1,177 T1LC-associated DEGs shared by both data sets, 110 genes with high expression levels in T1LCs (mean log2[TPM+1] > 4) but low expression levels in oogonia-like cells (adjusted log2[RPM+1] < 2) were selected. The expression values for these genes in the cell types in this study and cell types in Yamashiro et al. were shown in the heatmap.

**Profiling of TE expression levels from 10× chromium scRNA-seq data.** To quantify the expression level of TEs at locus resolution, we first prepared a transcript annotation file (i.e., GTF file) including human genes and TEs, annotated by RepeatMasker (Smit, AFA, Hubley, R & Green, P. *RepeatMasker Open*-4.0. 2013-2015; http://www.repeatmasker.org)[62]. Reads were mapped and counted using Cell Ranger with the above annotation file. TE loci detected in ≥0.5% of the cells were used in the downstream analyses. The read abundances of total TEs, respective TE families, and TE subfamilies were calculated by summing the read counts of TE loci. The expression levels of TEs were normalized as log$_2$-transformed counts per 10,000, with a pseudocount of 1 (log$_2$[CP10k + 1]), using Seurat (3.1.4)[63].

Dimension reduction analysis of the TE expression profile obtained from scRNA-seq data were performed according to the Seurat flamework (https://satijalab.org/seurat/). The expression level of TEs was normalized by sctransform[64] and the 100 most variably expressed TE subfamilies were selected using the FindVariableFeatures command. After the data were scaled and whitened, dimension reduction analysis was performed using the Uniform Manifold Approximation and Projection (UMAP)[65]. The first ten principal components were used in the analysis.

**Epigenetic profiling of TEs.** ATAC-seq data were downloaded from the NCBI Sequence Read Archive (SRA) using SRA Toolkit (2.10.2) and mapped to the human reference genome (GRCh38-3.0.0) using BWA Mem (0.7.17) with the default parameters[66]. Duplicated reads were removed using Picard MarkDuplicates (2.18.6) (https://broadinstitute.github.io/picard/). ATAC-seq peaks were called using MACS2 (2.2.6) with the default parameters[67]. The top 50,000 called peaks were used for downstream analysis. Fold-enrichment values for the overlaps between ATAC-seq peaks and a specific subfamily of TEs was calculated using an in-house script (calc_enrichment_randomized.py).

**Statistics and reproducibility.** Statistical analysis was performed in Excel and RStudio (version 1.2.5019, http://www.rproject.org/). Statistical significance of differences was determined as stated in the figure legend or elsewhere in the methods section. All fluorescence and histologic images and flow cytometric data were representatives of at least three independent experiments with similar results obtained at each experiment, unless stated otherwise in figures.

**Reporting summary.** Further information on research design is available in the Nature Research Reporting Summary linked to this article.

## Data availability
The datasets generated in this study as well as published data are available at NCBI GEO under the following accession numbers: the scRNA-seq data generated in this study "GSE153819"; the scRNA-seq data of migrating, mitotic and mitotic-arrest FGCs "GSE86146"[11] and of neonatal prospermatogonia "GSE124263"[18]; and the RNA-seq data of hPGCLCs and xrOvaries "GSE117101"[15]. Human infertility genes can be retrieved from Online Inheritance in Man (OMIM) at https://www.omim.org. 9A13 hiPSC lines are available from the corresponding authors upon request. Tissues, sections or cDNA derived from embryos are not available due to restriction in our study protocol. A reporting summary for this Article is available as a Supplementary Information file. Raw data associated with each figure are provided in the Source Data table of this paper. The remaining data are available within the Article, Supplementary Information, or available from the author upon request. The source data underlying Figs. 1d, j, 3e, j, k, and Supplementary Figs. 2c, d, 3c, d are provided as a Source Data file with this paper. Source data are provided with this paper.

## Code availability
Code for an in-house script (calc_enrichment_randomized.py) is in the GitHub repository at https://doi.org/10.5281/zenodo.4061019.

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

## Acknowledgements

We thank Drs. Leslie King, Kenneth Zaret, and Jeremy Wang for carefully reviewing the manuscript and providing insightful comments. We thank members of Sasaki lab for the discussion of this study and Ms. Karen Makar for her technical and administrative assistance. We thank Dr. Katalin Susztak for letting us use 10× Genomics Chromium Controller. We acknowledge the Comparative Pathology Core at the University of Pennsylvania School of Veterinary Medicine for the preparation of paraffin sections. This work was supported in part by Open Philanthropy fund from Silicon Valley Community Foundation (2019-197906) to K. Sasaki, and NIH R01 HD090007 to B.P.H. This research was supported by Basic Science Research Program through the National Research Foundation of Korea (NRF) funded by the Ministry of Education (NRF-2019R1A6A3A03032063) to Y.S.H.

## Author contributions

K. Sasaki and Y.S.H. conceived the project, designed the experiments and wrote the manuscript. B.H. edited the manuscript. Y.S.H. conducted hPGCLC induction, xrTestis culture and analyses. K. Sasaki and Y. Seita collected in vivo samples and prepared for scRNA-seq. Y. Sakata., Y. Seita and K. Sasaki assisted in hPGCLC induction and preparation for xrTestis culture. K. Sasaki, S.S., B.P.H, J.I., H.S., and K. Sato. contributed to the analyses of scRNA-seq.

## Competing interests

The authors declare no competing interests.
