## [Peer Review File · Nature Communications]

Reviewers' Comments:

Reviewer #1:

Remarks to the Author:

This study reveals molecular features and a linear trajectory of human prospermatogonia in human fetal testes. Remarkably, the authors reconstituted the induction of prospermatogonia-like cells from human induced pluripotent stem cells in vitro using xenogenic reconstituted testes. Their results showed gene expression profiles very similar to in vivo prospermatogonia. As the molecular features of human fetal germ cells are enigmatic and not accessible, the reconstitution of human prospermatogonia from iPSC cells (hiPSCs) is a powerful method for basic research and for the future development of potential therapeutic methods. Overall, this is an important work supported by innovative technology. I have several questions and suggestions to improve the quality of the study, and I enthusiastically support the publication of this work.

1. The definition of prospermatogonia (gonocytes) was initially introduced in mouse studies. Interestingly, this study and some precedent studies revealed that human fetal germ cells are highly heterogeneous and not synchronized compared to mouse germ cells. This study refers to all pregonadal phase as PGCs in humans and later gonadal phase as prospermatogonia. However, in mice, PGCs are generally defined as a largely-common gonadal phase between males and females (until E13.5), and after sexual dimorphism, male germ cells are defined as prospermatogonia. As "M spermatogonia" show the feature of DNA demethylation, "M spermatogonia" appear to correspond to E13.5 PGCs or earlier in mice. The definition may be slightly misaligned. This point can be clarified in writing.
2. Another remarkable feature of reconstituted human prospermatogonia is the slow progression from hPGCLC to T1 prospermatogonia (~ 120 days in culture). Is the progression (PGCs to T1) typical for in vivo human spermatogenesis? In mice, it takes only a few days from PGCs to T1 prospermatogonia. This timeline can be clarified in the manuscript and can be shown in the schematic in Fig 6h.
3. Reconstituting xenogeneic testes is an innovative method. These testes appear to lack Leydig cells. When do Leydig cells function in human spermatogenesis? How long can male germ cells survive and progress after 120 days in culture?
4. DNA demethylation, interestingly, takes place in MLC (Figure S3D). What is the DNA methylation level in T1LC?
5. In the reconstituted testes, transitional cells (TCs) were identified. Interestingly, a marker of TCs, ASB, was expressed in between M and T1 cells in vivo. Are TCs defined as a subpopulation in germ cell development in vivo?
6. T1LCs are compared with oogonia-like cells. Interestingly, MLCs and M prospermatogonia are similar to the "mitotic arrest" phase of female reconstituted germ cells. Do they correspond to "ground-state" PGCs in both males and females?

Minor points.

1. The study uses many abbreviations without clear definitions. "M" and "T1" prospermatogonia should be explained when they are introduced. "T1LSs" and "iMeLCs" should be defined as well.
2. Fig. S1a: please show examples of "large vesicular nuclei and prominent nucleoli."
3. P14. "Pair-wise DEGs between hPGCLCs and MLCs" can be shown, or alternatively delete this phase.
4. Hs26 and Hs27 (cryopreserved- thawed) and Hs31 (fresh) can be clearly explained in the method section.
5. Discussion: "Derivation of functional male GCs from hiPSCs was recently accomplished by several groups in mice". Since these studies did not use human cells, "hiPSCs" should be "iPSCs."
6. Figure panels 2h and 3c are very clouded and difficult to follow. They can be subdivided.

Reviewer #2:

Remarks to the Author:

Hwang YS., et al used scRNA-seq to investigate the cellular subpopulations during human male germline development. It is an interesting study. The authors did a very solid and comprehensive study on both experimental and computational aspects.

Major comments:

Emerging evidence suggest that the cell dissociation can substantially affect subpopulations in a single-cell RNA-seq.

First, certain cellular subpopulations can be missed at the cell dissociation step. The authors observed an increase of certain subpopulations (Page 14). This could also be due to a cell dissociation bias. I suggest in addition to compare % of cells in a subpopulation (between conditions), a comparison for the total number of cells (after cell dissociations) changes can be useful to rule out a potential cell dissociation bias.

Second, a study comparing different scRNA-seq datasets suggest that certain subpopulations were existing in multiple unrelated tissues (<https://www.nature.com/articles/nmeth.4437>). Then they have shown that these "subpopulations" were in fact due to certain genes being differentially expressed before and after cell dissociation (nothing to do with any biological function). I suggest authors to compare these "cell dissociation sensitive genes" with their DEG gene list as well as subpopulations to rule out a potential bias.

Minor comments:

It is unclear what is the background of the pathway/GO analysis. The enriched GO or pathways can be affected by different background list.

Reviewer #3:

Remarks to the Author:

Using detailed transcriptome profiling and decrease proliferation as criteria, Hwang et al., report reconstitution of prospermatogonia from human induced pluripotent stem cells. Germ and somatic cells were clustered using previously annotated marker transcripts including mitotic (M), migrating (PGC) and post-arrest (T1) FGS. Overall lineal trajectory from M to T1 cells was confirmed. Using a triple KI fluorescence reporter hiPSCs clonal cell line, the authors documented that the human cells self-assemble with dissociated mouse testicular somatic cells to form min-fetal testicular tissues (xrTestis) that mature in culture and became T1-like cells (T1LC), a process that corresponded to in vivo maturation based on scRNA-seq. In comparison with the freshly prepared Hs31 sample, the authors observe similar progression of fetal germline development in vitro and in vivo. Lastly, the manuscript compares expression dynamics of transposable elements in in vitro and in vivo samples.

The authors provide a well-written, exhaustively detailed manuscript with a thoughtful discussion of the advantages and disadvantages of in vitro xrTestis cultures. Although they suggest future improvements, it remains to be determined if the T1LCs can form spermatogonia stem cells for recapitulation of spermatogenesis.

Minor points:

1. Should "the perinuclear regions of MAGEC2+ M (Fig. 1g)" on page 8 be "MAGEC2+ T"?
2. Please explain abbreviations the first time that they are used in the manuscript, i.e., iMeLCs, tSNE and so on.
3. The "pair-wise DEGs between hPGCLCs and MLCs (data not shown)" on page 14 should be included in supplementary results.

4. Should "either whole GCs or T1LCs" on page 17 be "GCs or T1"?
5. Should "when compared with M or T1, respectively (Fig. 4c)" on page 17 be "Fig. 5c"?
6. The manuscript seems to lack information about "ag77 AG+VT-".
7. Information on antibodies (source, concentrations, etc.) needs to be included in Methods or in a supplementary file.
8. Should the descriptor "Supplementary Fig. 1-7" in supplementary data be "Supplementary Fig. 1-6" to correspond to the number of figures?
9. In general, the manuscript is wordy and would benefit from careful copy-editing.

Reviewer #4:

Remarks to the Author:

The transition process from PGC to prospermatogonia (gonocytes) then to SSCs population remains largely unknown, especially in humans. In this manuscript, Hwang et, al. uncovered human PGCs to gonocytes transition process by using scRNA Seq with human embryonic testicular tissues (in vivo) and fresh established in vitro differentiation model and showed that xenogeneic reconstituted testis could be a well model to study the early events of human germ cell development. Their work is meaningful and practical, the manuscript is well organized and should be acceptable when the following concerns are addressed.

- 1) The authors proposed that transitional cells (TCs) exist between in vitro M-T1 transition and claimed strong ASB expression cells were also found in the scRNA data set from testicular tissues. Whether strong ASB9 expression cells in vivo also express M and T1 markers, like POU5F1, Nanos3, Fox15. If TCs also exist in vivo, authors should point this out and change TCs in vitro as TC like cells;
- 2) In page 21, the first paragraph, "In summary..." From the TEs expression data, it seems that ERVs rather than TEs are the ones which may function as regulatory elements, authors should rephrase this paragraph;
- 3) In page 22, the first paragraph, authors referred to the wrong figure;
- 4) In figure legends Fig6g, references should also be added in the text;
- 5) In Fig1 panel d, "MKI67" need to be capital;
- 6) Better to reorganize Figure 1, it is difficult to follow
- 7) In page 22, I am not clear what authors mean when they wrote "using the same hiPSC lines of both XX and XY karyotypes..."

Rebuttal: NCOMMS-20-25300

We would like to sincerely thank the reviewers for their constructive comments, which we have used as the basis for revising our manuscript.

Summary of the manuscript revisions

The major request made by the editor/reviewers in regard to our manuscript was to address the cell dissociation bias that might affect scRNA-seq outputs. This point, among others, has been addressed with additional data. In addition to changes in response to the reviewers' comments, we have made the following minor modifications to improve the quality of the paper.

1. In addition to TFAP2C, we have added DDX4 as a marker for germ cells in our IF analysis of rTestes to confirm the absence of endogenous mouse germ cells in a more stringent manner (Fig.2d in the revised manuscript).
2. IF for human mitochondrial antigen (hMito), a pan-human specific marker, has been added to confirm the human origin of the germ cells present in xrTestes at d14 (Fig.2d, i in the revised manuscript).
3. Reference section was streamlined by removing redundant references.

We have addressed the reviewers' specific comments below.

Reviewer's comments:

Reviewer #1

This study reveals molecular features and a linear trajectory of human prospermatogonia in human fetal testes. Remarkably, the authors reconstituted the induction of prospermatogonia-like cells from human induced pluripotent stem cells in vitro using xenogenic reconstituted testes. Their results showed gene expression profiles very similar to in vivo prospermatogonia. As the molecular features of human fetal germ cells are enigmatic and not accessible, the reconstitution of human prospermatogonia from iPS cells (hiPSCs) is a powerful method for basic research and for the future development of potential therapeutic methods. Overall, this is an important work supported by innovative technology. I have several questions and suggestions to improve the quality of the study, and I enthusiastically support the publication of this work.

Response 1. We would like to sincerely thank the reviewer for these encouraging comments on our manuscript.

1. The definition of prospermatogonia (gonocytes) was initially introduced in mouse studies. Interestingly, this study and some precedent studies revealed that human fetal germ cells are highly heterogeneous and not synchronized compared to mouse germ cells. This study refers to all pregonadal phase as PGCs in humans and later gonadal phase as prospermatogonia. However, in mice, PGCs are generally defined as a largely-common gonadal phase between males and females (until E13.5), and after sexual dimorphism, male germ cells are defined as prospermatogonia. As "M spermatogonia" show the

feature of DNA demethylation, "M spermatogonia" appear to correspond to E13.5 PGCs or earlier in mice. The definition may be slightly misaligned. This point can be clarified in writing.

Response 2. We are aware that in some studies, both male and female mouse germ cells are referred to as PGCs at early gonadal phase (until E13.5), owing to the less obvious sexual dimorphism in morphology and cellular properties^{1,2}. However, sexual dimorphism in mouse germ cells is already evident at the molecular level at E9.5 (pre-gonadal phase), with a differential expression pattern for X-linked genes³.

A previous study⁴ in humans has shown that male mitotic fetal germ cells (FGCs), which correspond to M prospermatogonia, exhibit differential expression of several sex chromosomal and autosomal genes, as compared with the expression in female counterparts at the corresponding stages. Importantly, according to this preceding study, M prospermatogonia have been observed as early as 4 to 5 weeks post fertilization (very early gonadal stage, corresponding to ~E10.5 in mice). These findings suggest that sexual dimorphism may exist in gonadal germ cells in their earliest stages.

The presence of a functional "ground state" in PGCs has been inferred in mice on the basis of observations using reconstitution cultures between XX PGCs and XY somatic cells, or XY PGCs and XX somatic cells. In these experiments, XX gonadal PGCs (before E12.5) can develop as prospermatogonia, and XY gonadal PGCs (before E11.5) can develop as oogonia^{2,5}. However, neither completes meiosis or develops functional gametes. Therefore, the presence of a functional "ground state" in PGCs in a strict sense, has not been verified, even in mice.

Nonetheless, at the earliest gonadal phase, both male and female GCs exhibit many similarities in cellular properties, as compared with those of later gonadal phases. How this sexual dimorphism is shaped warrants further investigation in both mice and humans. Our xrTestes in conjunction with xrOvaries from a preceding study now provide an excellent platform to characterize such sexual dimorphism during human germ cell development⁶. Given all these points, we consider it appropriate to adhere to the original nomenclature system for male germ cells introduced by Hilscher et al., 1974⁷. We have succinctly addressed these points in the revised manuscript (p3, INTRODUCTION, third paragraph; p25-26, DISCUSSION, seventh paragraph).

2. Another remarkable feature of reconstituted human prospermatogonia is the slow progression from hPGCLC to T1 prospermatogonia (~ 120 days in culture). Is the progression (PGCs to T1) typical for in vivo human spermatogenesis? In mice, it takes only a few days from PGCs to T1 prospermatogonia. This timeline can be clarified in the manuscript and can be shown in the schematic in Fig 6h.

Response 3. The exact time frame of human prospermatogonial development in vivo is unknown to date, owing to the heterogeneous nature of human gonadal germ cell development and the lack of tools to trace the fate of individual human germ cells. According to a previous study⁴, T1 emerges as early as 9 weeks post fertilization (63 days). Therefore, the emergence of T1LCs is slower than the fastest timeline of T1 development in vivo. However, given that T1 and M were simultaneously present in all our testicular samples at 17 to 18 weeks and in samples from a previous study up to 26 weeks⁴, some germ cells may not progress into T1 until the third trimester. In this regard, the developmental timeline of our T1LCs may still be within a physiological range. These points are discussed in the revised manuscript (p23,

DISCUSSION, third paragraph in the revised manuscript), and the estimated timeline of human male germ cell development has been added to the schematic in Fig.6h in the revised manuscript.

3. Reconstituting xenogeneic testes is an innovative method. These testes appear to lack Leydig cells. When do Leydig cells function in human spermatogenesis?

Response 4. The reviewer is asking when Leydig cells (or fetal Leydig cells) function in human male germ cell development. The production of androgen is thought to be a major function of Leydig/fetal Leydig cells⁸. In this regard, androgen production affects spermatogonial maintenance in humans but not their fetal development per se, on the basis of the observation of spermatogonia in patients with complete androgen insensitivity who have androgen receptor mutations^{9,10}. In mice, ACTIVIN A produced by fetal Leydig cells is essential for Sertoli cell proliferation and tubulogenesis, and conditional knockout of ACTIVIN A in fetal Leydig cells results in tubular dysgenesis⁸. Therefore, loss of fetal Leydig cells in xrTestes might also have contributed to the tubular dissolution observed after long-term culture of xrTestes in this study. We have added a passage explaining these points (p23-24, DISCUSSION, fourth paragraph in the revised manuscript).

How long can male germ cells survive and progress after 120 days in culture?

Response 5. We cultured xrTestes up to d134. These xrTestes still harbored germ cells (AG⁺VT⁻, AG⁺VT⁺, AG⁻VT⁺) albeit with a lower frequency than that at d120 (Fig.R1 for reviewers). We are interested in longer culture for further differentiation of TILCs. However, because the tubular dissolution is already evident at d120, we consider that further differentiation should be pursued by using different approaches rather than continuous culture of xrTestes beyond day 120 (p23-24, DISCUSSION, fourth paragraph in the revised manuscript). We will address this issue in our future studies.

4. DNA demethylation, interestingly, takes place in MLC (Figure S3D). What is the DNA methylation level in TILC?

Response 5. In response to the reviewer's suggestion, we have newly examined the global DNA methylation states of germ cells in d120 xrTestes. IF for 5-methylcytosine (5mC) revealed that the DNA methylation levels of DDX4⁺ germ cells in d120 xrTestes were still lower than those in somatic cells, thus suggesting that overt *de novo* DNA methylation might not have commenced yet (Fig.3k in the revised manuscript). However, subtle changes in global DNA methylation levels between d77 and d120 might not have been detectable by IF, owing to its relatively low sensitivity and quantification ability. Evaluation of DNA methylation in TILCs at higher resolution, such as through whole genome bisulfite sequencing, will be performed in a separate future study, which should provide important insights into the epigenetic reprogramming occurring in prospermatogonial specification in vitro.

In addition, although the DDX4 expression was much higher in T1LCs and TCs than in MLCs, it is not the most specific marker for T1LCs (Fig.3c,d, Fig.4c and Supplementary fig.3b). We were unable to co-stain for 5mC and more specific T1LC markers for the following technical reason. IF analysis of 5mC involves hydrochloride treatment for DNA denaturation, which destroys most of the protein epitopes recognized by antibodies. Therefore, only limited antibodies can be used in conjunction with anti-5mC antibodies. Unfortunately, in our hands, a signal for MAGEC2, a more specific marker for T1LCs, was not detectable by IF after hydrochloride treatment. MAGEA3 is a mouse monoclonal antibody that consequently, in IF, cannot be combined with anti-5mC antibody, which is also a monoclonal mouse antibody. Therefore, we used DDX4 as a germ cell marker, which preferentially marks T1LCs and TCs over MLCs at d120 xrTestes.

5. In the reconstituted testes, transitional cells (TCs) were identified. Interestingly, a marker of TCs, ASB, was expressed in between M and T1 cells in vivo. Are TCs defined as a subpopulation in germ cell development in vivo?

Response 6. We would like to thank the reviewer for pointing this out. In response to the reviewer's comments, we have further analyzed *ASB9*-expressing cells *in vivo* by comparing the transcriptomes of M and T1. We found that *ASB9*-expressing cells *in vivo* showed intermediate levels of gene expression for both M (*POU5F1* and *NANOS3*) and T1 markers (*MAGEC2*, *NANOS2* and *TEX15*). Other M and T1 markers at the genome-wide level also followed the same pattern, analogously to the relationship of TCs to MLCs and T1LCs. Nonetheless, we were unable to identify an *ASB9*-expressing population *in vivo* as a distinct cluster in our computational analysis, even with higher clustering resolution (K-means up to 10) (Fig.R2 for reviewers). This result might have been due the cellular heterogeneity and various other complexities present in the *in vivo* milieu, thus making the TC population less discernible *in vivo*. We have added a succinct explanation of this point to the revised manuscript (p14, "Lineage trajectory leading to T1LCs," second paragraph in the revised manuscript). Please also see **Response 2** for reviewer #4.

6. T1LCs are compared with oogonia-like cells. Interestingly, MLCs and M prospermatogonia are similar to the "mitotic arrest" phase of female reconstituted germ cells. Do they correspond to "ground-state" PGCs in both males and females?

Response 7. We noticed that labeling for "mitotic" and "mitotic arrest" FGCs at the bottom of Fig.5e in the original manuscript were inadvertently switched; the labeling has been corrected in the revised Fig.5e. To avoid confusion, we have also changed hFGC(M) to FGCs (male). We apologize for the error and confusion. Therefore, MLCs and M prospermatogonia resemble "mitotic" *in vivo* FGCs (male) but not the "mitotic arrest" phase of female reconstituted germ cells. We noted similarities between MLCs (and M) and ag77 AG⁺VT⁺ cells (female reconstituted germ cells by Yamashiro et al., 2018)⁶ in the expression of M/MLCs markers (Fig.R3 for reviewers). However, because MLCs (XY) in this study already expressed some Y chromosomal genes at the MSY (male-specific region of Y chromosome), whereas ag77 AG⁺VT⁺ cells (XX) did not (Fig.5h in the original/revised manuscript), sexual dimorphism already exists in a strict sense between MLCs and ag77 AG⁺VT⁺ cells. Together with the reasons outlined in **Response 2**, we consider that applying the concept of "ground-state" PGCs for MLCs in this study or for

ag77 AG⁺VT⁺ cells by Yamashiro et al., 2018⁶, would require caution. Please also see **Response 2** for reviewer #1 and **Response 6** for reviewer #3.

Minor points. 1. The study uses many abbreviations without clear definitions. "M" and "T1" prospermatogonia should be explained when they are introduced. "TILSs" and "iMeLCs" should be defined as well.

Response 8. We have explained these abbreviations when they are introduced in the revised manuscript.

2. Fig.S1a: please show examples of "large vesicular nuclei and prominent nucleoli."

Response 9. We have added additional images for germ cells and fetal Leydig cells at higher magnification in the revised manuscript (Supplementary Fig.1a).

3. P14. "Pair-wise DEGs between hPGCLCs and MLCs" can be shown, or alternatively delete this phase.

Response 10. We have added a box plot for the expression of pairwise DEGs between hPGCLCs and MLCs in two replicates of hPGCLCs (hPGCLCs_1, hPGCLCs_2) in the revised manuscript (Supplementary Fig.4j).

4. Hs26 and Hs27 (cryopreserved- thawed) and Hs31 (fresh) can be clearly explained in the method section.

Response 11. We have explained this point in the method section (page 28, "Human testis sample preparation," second paragraph in the revised manuscript).

5. Discussion: "Derivation of functional male GCs from hiPSCs was recently accomplished by several groups in mice". Since these studies did not use human cells, "hiPSCs" should be "iPSCs."

Response 12. This has been corrected in the revised manuscript.

6. Figure panels 2h and 3c are very clouded and difficult to follow. They can be subdivided.

Response 13. Fig.2h in the original manuscript has been subdivided into Fig.2i and j in the revised manuscript. Fig.3c in the original manuscript has been subdivided into Fig.3c,d,g in the revised manuscript.

Reviewer #2

Hwang YS., et al used scRNA-seq to investigate the cellular subpopulations during human male germline development. It is an interesting study. The authors did a very solid and comprehensive study on both experimental and computational aspects.

Major comments:

Emerging evidence suggest that the cell dissociation can substantially affect subpopulations in a single-cell RNA-seq.

First, certain cellular subpopulations can be missed at the cell dissociation step. The authors observed an increase of certain subpopulations (Page 14). This could also be due to a cell dissociation bias. I suggest in addition to compare % of cells in a subpopulation (between conditions), a comparison for the total number of cells (after cell dissociations) changes can be useful to rule out a potential cell dissociation bias.

Response 1. We have provided these data (Supplementary Fig.4g in the original manuscript and Supplementary Fig.4h in the revised manuscript) and have included a description (page 14 in the original manuscript) as one of many pieces of evidence supporting our hypothesis that lineage progression occurs from MLCs to T1LCs in xrTestis culture, which appears in the preceding sentence in the original manuscript (page 14, "Lineage trajectory leading to T1LCs," second paragraph: "This analysis confirmed that lineage progression..."). This hypothesis has already been supported by various pieces of evidence in this paper, including RNA velocity analysis (Fig.4b), IF analysis of histologic sections (Fig.2i, Fig.3c,d,e,g,i and Supplementary Fig.3b) and flow cytometry (Fig.3f,h).

We agree with the reviewer that dissociation can introduce bias through cell death of certain populations or incomplete dissociation resulting in cell clumps, which are usually removed from the downstream analyses. Although it is difficult to completely rule out dissociation bias, a comparison with the undissociated native tissues can rule out overt bias affecting the representation of subpopulations. In this regard, a proportion of germ cell subpopulations at d77 xrTestes assessed by IF on sections (TFAP2C+DDX4⁻, ~30%; TFAP2C+DDX4⁺, ~53%; TFAP2C-DDX4⁺, ~17%) (Supplementary Fig.3c in the revised manuscript) was found to be similar to that of xrTestes at a similar stage (d81), as assessed by flow cytometry (after dissociation) (TFAP2C(AG)⁺DDX4(VT)⁻, ~25%; TFAP2C(AG)⁺DDX4(VT)⁺, ~62%; TFAP2C(AG)⁻DDX4(VT)⁺, ~13%) (Fig.3f in the revised manuscript; the percentage among germ cells has been recalculated by using the percentage among all cells in the figure), thus suggesting that dissociation did not introduce overt bias in the proportion of subpopulations present in d77/d81 xrTestes. Comparison of IF and flow cytometry results of the proportion of MLCs (and a fraction of TCs) present in d77/d81 or d120/d124 xrTestes also revealed similar trends between these two methods. A fraction of POU5F1⁺/NANOG⁺ cells (corresponding to MLCs and a certain fraction of TCs) among all germ cell (TFAP2C⁺ or DDX4⁺) decreased from 94% at d77 to 8.3% at d120, as determined by IF (Fig.3e in the revised manuscript), whereas AG⁺ cells (corresponding to MLCs and a fraction of TCs) among all germ cells (AG⁺ or VT⁺) also decreased from 90% at d77 to 41% at d120, as determined by flow cytometry

(Fig.3f in the revised manuscript). The over-representation of the proportion of MLCs (and some TCs) at d120 in flow cytometry (41%) compared with IF (8.3%) might have been due to dissociation bias, lot-to-lot variation or differences in the sensitivity for detecting germ cells by IF for POU5F1/NANOG versus by flow cytometry for AG. Despite some differences in the proportions, the overall trends remained the same: the proportion of MLCs declined as the culture extended from d77/81 to d120/124.

Also, of note, we did not identify doublets in our FACS analysis (Fig.R4 for reviewers), thus suggesting complete dissociation. Unfortunately, the total number of cells after dissociation of xrTestes were ~3000/aggregate and ~1000/aggregate for d77/81 and d120/124, respectively, a result that may have been due to loss of both somatic and germ cells at d120/124, at least in part because of tubular dissolution (please see DISCUSSION in the original and revised manuscript). Therefore, the total number of cells may not be particularly useful to rule out a potential dissociation bias.

Overall, dissociation bias might exist, but we do not consider it an impediment to our conclusions about MLC-to-T1LC transition along the time course.

Second, a study comparing different scRNA-seq datasets suggest that certain subpopulations were existing in multiple unrelated tissues (<https://www.nature.com/articles/nmeth.4437>). Then they have shown that these “subpopulations” were in fact due to certain genes being differentially expressed before and after cell dissociation (nothing to do with any biological function). I suggest authors to compare these “cell dissociation sensitive genes” with their DEG gene list as well as subpopulations to rule out a potential bias.

Response 2. We would like to thank the reviewer for pointing out this important technical issue. All cell clusters in this study were carefully annotated on the basis of the expression of known marker genes previously reported to mark the same or similar cell types. For example, clusters for SCs and ECs were annotated on the basis of the expression of known SC markers (*SOX9* and *INHBB*) and EC markers (*KDR* and *CLDN5*), respectively. T1LCs and MLCs were annotated according to the similarities in their gene expression patterns to T1 and M in fetal testes, respectively. TCs were annotated on the basis of position in tSNE space as well as intermediate levels of expression of both M and T1 markers. Moreover, key markers for T1/T1LCs and M/MLCs were also validated through other methods, such as immunofluorescence staining, in-situ hybridization or flow cytometry (Fig.1e,h,i, Fig.3b-i and Supplementary Fig.3b in the revised manuscript). We therefore believe that the clusters annotated and characterized in this study represent biologically meaningful cell populations rather than artifacts.

Nonetheless, in response to the reviewer 's comment, we have examined the expression of dissociation-induced genes in the cell types defined in this study. These genes were found to be expressed at various levels in multiple cell types, thus suggesting that dissociation may have affected gene expression in our cell types. Some of these dissociation-induced genes (1–4 out of 30) were found in our list of DEGs characteristic of each cell type; however, these were generally low in rank (sorted by FDR value in ascending order). Therefore, we concluded that overt enrichment of dissociation-induced genes in certain cell types was not seen in this study. We have incorporated the data obtained and the relevant description/discussion into the revised manuscript (Supplementary Fig.5a-c, page 14-15, "Lineage trajectory leading to T1LCs," fourth paragraph).

Minor comments:

It is unclear what is the background of the pathway/GO analysis. The enriched GO or pathways can be affected by different background list.

Response 3. The background for pathway/GO analysis in this study is "Homo sapiens." This point has been added in the revised manuscript (page 37-38, METHODS, "Mapping reads of 10X Chromium scRNA-seq and data analysis," second paragraph)

Reviewer #3

The authors provide a well-written, exhaustively detailed manuscript with a thoughtful discussion of the advantages and disadvantages of in vitro xrTestis cultures. Although they suggest future improvements, it remains to be determined if the TILCs can form spermatogonia stem cells for recapitulation of spermatogenesis.

Minor points:

1. Should "the perinuclear regions of MAGEC2+ M (Fig. 1g)" on page 8 be "MAGEC2+ T"?

Response 1. We apologize for the typographical error. It has been corrected accordingly.

2. Please explain abbreviations the first time that they are used in the manuscript, i.e., iMeLCs, tSNE and so on.

Response 2. We have explained abbreviations when they are introduced.

3. The "pair-wise DEGs between hPGCLCs and MLCs (data not shown)" on page 14 should be included in supplementary results.

Response 3. We have incorporated the data (Supplementary Fig.4j in the revised manuscript).

4. Should "either whole GCs or TILCs" on page 17 be "GCs or T1"?

Response 4. This has been corrected accordingly.

5. Should "when compared with M or T1, respectively (Fig.4c)" on page 17 be "Fig.5c"?

Response 5. This has been corrected accordingly.

6. *The manuscript seems to lack information about “ag77 AG+VT-”.*

Response 6. We noticed that and "ag 77 AG⁺VT⁺ cells" were inadvertently referred to as "ag 77 AG⁺VT⁻ cells" in the original manuscript. This has been corrected to "ag 77 AG⁺VT⁺ cells" in the revised manuscript. We apologize for the error. We have added a succinct explanation regarding these cells and ag120 AG⁺VT⁺ cells from Yamashiro et al., 2018⁶ (page 19, "Comparison of TILCs with oogonia-like cells induced from hiPSCs," first paragraph).

7. *Information on antibodies (source, concentrations, etc.) needs to be included in Methods or in a supplementary file.*

Response 7. We have added information about antibodies in Supplementary Table 7 in the revised manuscript.

8. *Should the descriptor “Supplementary Fig.1-7” in supplementary data be “Supplementary Fig.1-6” to correspond to the number of figures?*

Response 8. We have newly added two Supplementary figures. Therefore, this has been corrected to “Supplementary Fig.1–8.”

9. *In general, the manuscript is wordy and would benefit from careful copy-editing.*

Response 9. We have streamlined and copy-edited the manuscript.

Reviewer #4

Their work is meaningful and practical, the manuscript is well organized and should be acceptable when the following concerns are addressed.

Response 1. We sincerely thank the reviewer for this encouraging comment.

1) The authors proposed that transitional cells (TCs) exist between in vitro M-T1 transition and claimed strong ASB expression cells were also found in the scRNA data set from testicular tissues. Whether strong ASB9 expression cells in vivo also express M and T1 markers, like POU5F1, Nanos3, Fox15. If TCs also exist in vivo, authors should point this out and change TCs in vitro as TC like cells;

Response 2. Please see **Response 6** for reviewer #1.

2) In page 21, the first paragraph, “In summary...” From the TEs expression data, it seems that ERVs rather than TEs are the ones which may function as regulatory elements, authors should rephrase this paragraph;

Response 3. We have corrected this paragraph accordingly (page 21, “Expression dynamics of transposable elements during human male germline development,” fourth paragraph in the revised manuscript).

3) In page 22, the first paragraph, authors referred to the wrong figure;

Response 4. This has been corrected to Fig.6h (page 22, DISCUSSION, first paragraph in the revised manuscript).

4) In figure legends Fig6g, references should also be added in the text;

Response 5. References have been added (Fig.6g, page 21, “Expression dynamics of transposable elements during human male germline development,” third paragraph in the revised manuscript).

5) In Fig1 panel d, “MKI67” need to be capital;

Response 6. This has been corrected accordingly.

6) Better to reorganize Figure 1, it is difficult to follow

Response 7. This has been reorganized in the revised manuscript.

7) In page 22, I am not clear what authors mean when they wrote “using the same hiPSC lines of both XX and XY karyotypes...”

Response 8. We apologize for the unclearness. We removed “of both XX and XY karyotypes” from this sentence (page 23, DISCUSSION, second paragraph).

References

- 1 McLaren, A. & Southee, D. Entry of mouse embryonic germ cells into meiosis. *Dev Biol* **187**, 107-113, doi:10.1006/dbio.1997.8584 (1997).
- 2 Adams, I. R. & McLaren, A. Sexually dimorphic development of mouse primordial germ cells: switching from oogenesis to spermatogenesis. *Development* **129**, 1155-1164 (2002).
- 3 Sangrithi, M. N. *et al.* Non-Canonical and Sexually Dimorphic X Dosage Compensation States in the Mouse and Human Germline. *Dev Cell* **40**, 289-301 e283, doi:10.1016/j.devcel.2016.12.023 (2017).
- 4 Li, L. *et al.* Single-Cell RNA-Seq Analysis Maps Development of Human Germline Cells and Gonadal Niche Interactions. *Cell Stem Cell* **20**, 858-873 e854, doi:10.1016/j.stem.2017.03.007 (2017).
- 5 Taketo, T. The role of sex chromosomes in mammalian germ cell differentiation: can the germ cells carrying X and Y chromosomes differentiate into fertile oocytes? *Asian J Androl* **17**, 360-366, doi:10.4103/1008-682X.143306 (2015).
- 6 Yamashiro, C. *et al.* Generation of human oogonia from induced pluripotent stem cells in vitro. *Science* **362**, 356-360, doi:10.1126/science.aat1674 (2018).
- 7 Hilscher, B. *et al.* Kinetics of gametogenesis. I. Comparative histological and autoradiographic studies of oocytes and transitional prospermatogonia during oogenesis and prespermatogenesis. *Cell Tissue Res* **154**, 443-470, doi:10.1007/BF00219667 (1974).
- 8 Wen, Q., Cheng, C. Y. & Liu, Y. X. Development, function and fate of fetal Leydig cells. *Semin Cell Dev Biol* **59**, 89-98, doi:10.1016/j.semcdb.2016.03.003 (2016).
- 9 Hannema, S. E. *et al.* Testicular development in the complete androgen insensitivity syndrome. *J Pathol* **208**, 518-527, doi:10.1002/path.1890 (2006).
- 10 Kaprova-Pleskacova, J. *et al.* Complete androgen insensitivity syndrome: factors influencing gonadal histology including germ cell pathology. *Mod Pathol* **27**, 721-730, doi:10.1038/modpathol.2013.193 (2014).

Reviewers' Comments:

Reviewer #1:

Remarks to the Author:

The authors fully addressed my previous suggestions. I found that the manuscript was substantially improved after the revision. I believe that this will be an impactful and influential work.

One minor point that was not clarified is the issue of abbreviation; M-prospermatogonia is "mitotic" prospermatogonia, and T1-prospermatogonia is "Transitional 1" prospermatogonia. These definitions should be added in the introduction.

Satoshi Namekawa

Reviewer #2:

Remarks to the Author:

The authors have addressed all of my questions/concerns. It is a nice work indeed.

Reviewer #3:

Remarks to the Author:

My concerns have been addressed in the revised manuscript.

Reviewer #4:

Remarks to the Author:

Authors addressed my concerns well. From my point of view, this manuscript is ready for publication.

Rebuttal: NCOMMS-20-25300A

We would like to sincerely thank the reviewers for their constructive comments, which we have used as the basis for revising our manuscript.

We have addressed the reviewers' specific comments below.

Reviewer's comments:

Reviewer #1

The authors fully addressed my previous suggestions. I found that the manuscript was substantially improved after the revision. I believe that this will be an impactful and influential work.

One minor point that was not clarified is the issue of abbreviation; M-prospermatogonia is "mitotic" prospermatogonia, and T1-prospermatogonia is "Transitional 1" prospermatogonia. These definitions should be added in the introduction.

Satoshi Namekawa

Response 1. We would like to sincerely thank the reviewer for these encouraging comments on our manuscript. We added definitions for M-prospermatogonia and T1-prospermatogonia in the revised manuscript (p3, INTRODUCTION, third paragraph in the revised manuscript)